# The thioredoxin-1 system is essential for fueling DNA synthesis during T-cell metabolic reprogramming and proliferation

Jonathan Muri [1], Sebastian Heer[1], Mai Matsushita[1], Lea Pohlmeier[1], Luigi Tortola[1], Tobias Fuhrer [2], Marcus Conrad[3], Nicola Zamboni[2], Jan Kisielow[1] & Manfred Kopf [1]

The thioredoxin-1 (Trx1) system is an important contributor to cellular redox balance and is a sensor of energy and glucose metabolism. Here we show critical c-Myc-dependent activation of the Trx1 system during thymocyte and peripheral T-cell proliferation, but repression during T-cell quiescence. Deletion of thioredoxin reductase-1 (*Txnrd1*) prevents expansion the CD4$^-$CD8$^-$ thymocyte population, whereas *Txnrd1* deletion in CD4$^+$CD8$^+$ thymocytes does not affect further maturation and peripheral homeostasis of αβT cells. However, *Txnrd1* is critical for expansion of the activated T-cell population during viral and parasite infection. Metabolomics show that TrXR1 is essential for the last step of nucleotide biosynthesis by donating reducing equivalents to ribonucleotide reductase. Impaired availability of 2′-deoxyribonucleotides induces the DNA damage response and cell cycle arrest of *Txnrd1*-deficient T cells. These results uncover a pivotal function of the Trx1 system in metabolic reprogramming of thymic and peripheral T cells and provide a rationale for targeting Txnrd1 in T-cell leukemia.

[1] Institute of Molecular Health Sciences, ETH Zurich, 8093 Zürich, Switzerland. [2] Institute of Molecular Systems Biology, ETH Zurich, 8093 Zurich, Switzerland. [3] Institute of Developmental Genetics, Helmholtz Center, Neuherberg 85764, Germany. These authors contributed equally: Jonathan Muri, Sebastian Heer. Correspondence and requests for materials should be addressed to M.K. (email: Manfred.Kopf@ethz.ch)

Aerobic organisms have developed a complex system of antioxidant proteins that maintains a reducing environment in the cell. The thioredoxin (Trx) system is one of the major cellular disulfide reducing systems comprising NADPH, Trx reductase (TrxR, encoded by *Txnrd*), and Trx (encoded by *Txn*). TrxR possesses the unique capacity to utilize electrons from NADPH to recycle oxidized Trx to its reduced form[1,2]. The Trx system regulates multiple cellular processes, such as gene expression, antioxidant response, apoptosis, and proliferation. In particular, it acts as electron donor for the antioxidant enzymes peroxiredoxins and methionine sulfoxide reductases[3,4], modulates the activity of transcription factors such as NF-κB, Ref1, and HIF1α[5–7], and is involved in the regulation of apoptosis by suppressing ASK1 function[8]. In mammals, the glutathione (GSH)–glutaredoxin (Grx) system has many overlapping activities with the Trx system due to extensive crosstalk[9]. For instance, Trx and Grx redundantly deliver reducing equivalents for the enzymatic activity of ribonucleotide reductase (RNR)[10]. However, how antioxidant systems regulate immune cell function and to which extent the Trx and Grx systems can compensate for each other functions in vivo are poorly understood at present.

Mammals express three TrxR isozymes that differ in their intracellular localization and tissue-specific expression pattern: TrxR1 localized in the cytoplasm, TrxR2 localized in mitochondria, and TrxR3 specifically expressed in the testis. Moreover, Trx1 and Trx2 are isoforms localized in the cytoplasm and in mitochondria respectively[9]. This implicates distinct roles at different cellular compartments. Thioredoxin-interacting protein (Txnip) binds to reduced Trx and acts as a negative regulator of Trx functions[11]. *Txnip* knockout in mice results in hypoglycemia, hyperinsulinemia, and liver steatosis[12]. Moreover, disruption of Txnip in obese mice strikingly improves hyperglycemia and glucose intolerance[13], demonstrating a crucial role of Txnip in metabolic disorders. Independently of its Trx-binding function, Txnip represses cellular glucose uptake by both reducing *Glut1* expression and inducing Glut1 internalization[14–17]. *Txnip* expression is mainly mediated by the glucose-sensing transcription complexes chREBP–Mlx and MondoA–Mlx, which bind to carbohydrate response element on the *Txnip* promoter[18,19].

Due to the critical functions of Txnip in regulating glucose metabolism, we hypothesized the Txnip–Trx system might play a role in the metabolic changes occurring upon T-cell stimulation. Notably, in contrast to naive T cells, activated T cells consume large amount of glucose and amino acids, thereby adjusting their metabolism toward increased glycolysis and glutaminolysis[20,21]. Previously, the mitochondrial Trx system was described to be dispensable for development, maintenance, and proliferation of lymphocytes[22]. To establish a potential function the cytosolic Trx system in T-cell-mediated immunity and metabolism, we generated *Txnrd1*[fl/fl];*Cd4-Cre* αβ T-cell-specific and tamoxifen (TAM)-inducible (*Txnrd1*[fl/fl];*Cre-ERT2*) mice. We found that T-cell activation is linked to a drastic downregulation of *Txnip* and an increase in *Txn1/Txnrd1* expression, which is absolutely required for synthesis of 2′-deoxyribonucleotides during T-cell metabolic reprogramming. These results consequently characterized a previously unknown function of the cytosolic Trx system in T-cell development and responses.

## Results

***Txnrd1* is essential for thymic iNKT cell development.** To investigate the function of the Trx system in T cells, we generated *Txnrd1*[fl/fl];*Cd4-Cre* mice by crossing mice with loxP-flanked *Txnrd1* alleles to mice expressing Cre recombinase from the *Cd4*

promoter. In these mice, deletion of *Txnrd1* mainly occurs in CD4[+]CD8[+] double positive (DP) thymocytes, and consequently both CD4[+] and CD8[+] αβ T cells and CD1d-resticted, invariant natural killer T (iNKT) cells lack *Txnrd1*. Cre-mediated deletion in DP and mature thymocytes, and in peripheral CD4[+] and CD8[+] T cells from *Txnrd1*[fl/fl];*Cd4-Cre* mice was complete at the genomic DNA and mRNA levels (Supplementary Fig. 1a,b). Wild-type (WT) and *Txnrd1*[fl/fl];*Cd4-Cre* mice showed comparable frequencies and numbers of thymic populations of CD4[−]CD8[−] double-negative (DN), CD4[+]CD8[+] DP and CD4[+] and CD8[+] single-positive (SP) T cells (Fig. 1a). Furthermore, *Txnrd1* deficiency had no effects on peripheral αβ T cell numbers in spleen, lymph nodes (LNs), and liver (Fig. 1b and Supplementary Fig. 1c). Expectedly, a proportion of peripheral CD4[+] and CD8[+] T cell in naive WT mice displayed an activated/memory phenotype (i.e., CD62L[hi]CD44[hi] and CD62L[lo]CD44[hi]). However, *Txnrd1*[fl/fl];*Cd4-Cre* mice had a considerably lower percentage of such cells in the spleen, LNs, and the liver (Fig. 1c and Supplementary Fig. 1d).

We next assessed the ability of *Txnrd1*-deficient T cells to refill the hematopoietic compartment of lethally irradiated hosts in a competitive situation with WT cells. Therefore, we reconstituted irradiated C57BL/6 mice with an equal ratio of congenically marked donor bone marrow (BM) cells from *Txnrd1*[fl/fl];*Cd4-Cre* (CD45.2[+]) and WT (CD45.1[+]) mice. In this setting, *Txnrd1*-deficient CD4[+] and CD8[+] T cells had a clear disadvantage compared to WT cells in peripheral organs but not in the thymus (Fig. 1d and Supplementary Fig. 1e). This suggests that *Txnrd1* is required to refill the peripheral hematopoietic compartment but not for thymic selection and maturation. In line with the low number of activated/memory T cells in *Txnrd1*-deficient naive mice (Fig. 1c and Supplementary Fig. 1d), the defect in contribution of *Txnrd1*-deficient T cells to this compartment was even more pronounced in a competitive setting (Fig. 1e and Supplementary Fig. 1f). Taken together, these data suggest that *Txnrd1* is dispensable for selection of conventional αβ DP T cells in the thymus and their homeostasis in the periphery. Moreover, *Txnrd1* is required intrinsically for expansion of T cells in a lymphopenic environment and steady state generation of activated/memory T cells.

In contrast to conventional αβ T cells, we found that *Txnrd1*-deficient iNKT cells were massively reduced in the thymus, liver, and spleen (Fig. 1f), suggesting a crucial function of *Txnrd1* in iNKT cell development. iNKT cells are known to arise from DP T cells and undergo massive thymic expansion thereafter[23]. In the absence of *Txnrd1*, thymic CD1d-PBS57-tetramer[+]PLZF[+] iNKT cell numbers were unaffected at stage 0 (CD24[hi]CD44[lo]NK1.1[−]) but were reduced at stages 1 (CD24[lo]CD44[lo]NK1.1[−]), 2 (CD24[lo]CD44[hi]NK1.1[−]), and 3 (CD24[lo]CD44[hi]NK1.1[+]), indicating a defect in expansion rather than failed thymic selection (Fig. 1g). In contrast to thymic iNKT cells, expansion of T cell precursors occurs mainly at the DN stage, and therefore prior *Txnrd1* deletion in *Txnrd1*[fl/fl];*Cd4-Cre* mice. Interestingly, by examining expression of the three main components of the Trx1 system including TrxR1, Trx1, and Txnip, the inhibitor of Trx1, we found that both *Txn1* and *Txnrd1* expression was increased in DN compared to DP and SP T cells. In contrast, *Txnip* was predominantly expressed in DP and SP compared to DN T cells (Supplementary Fig. 1g). DN precursors undergo two separate phases of proliferation. Pre-β proliferation occurs prior to TCR β-chain rearrangement at the DN2 stage followed by pre-α proliferation before TCRα chain rearrangement at the DN4 stage[24]. Precursor proliferation ceases at a stage characterized as CD4[+]CD8[+]TCR[−/lo]FCS[hi] (DP blasts) and it comes to a rest at the CD4[+]CD8[+]TCR[−/lo]FCS[lo] stage (DP rest). To better resolve the expression profiles of the Trx system components

during thymic development, we FACS-sorted distinct thymocyte populations. We consistently observed that phases of precursor proliferation (i.e., DN2 and DN4) were associated with low levels of *Txnip* but high levels of both *Txn1* and *Txnrd1*. By contrast, cells in resting phases (i.e., DN1, DN3, DP, and SP) displayed high *Txnip*, but low *Txn1* and *Txnrd1* expression (Fig. 1h and Supplementary Fig. 1h). These data indicate that cell proliferation may be licensed by a high *Txnrd1–Txn1:Txnip* ratio and that TrxR1 may play a role for expansion of DN precursors, which would be missed in *Txnrd1*$^{fl/fl}$;*Cd4-Cre* mice.

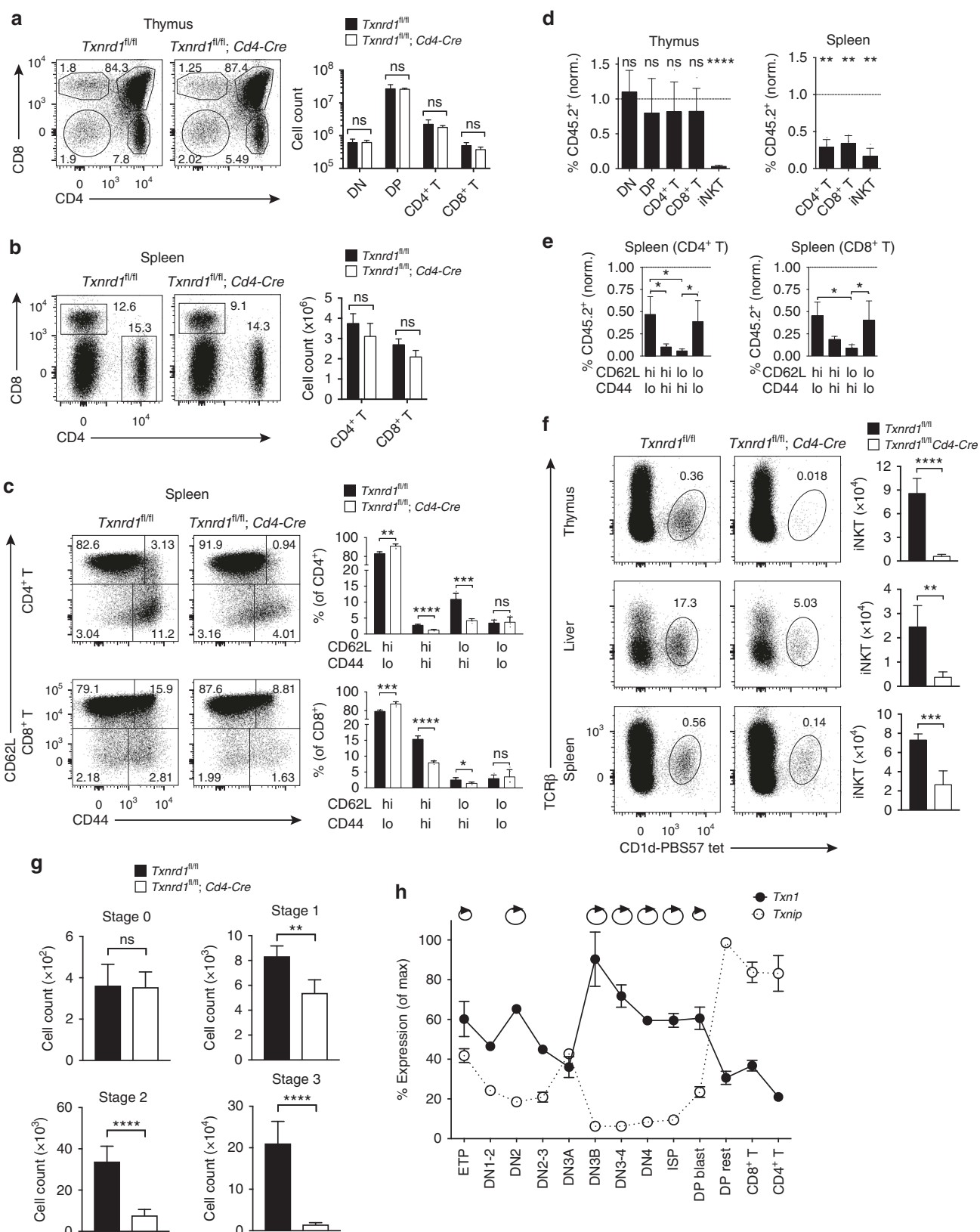

**Txnrd1 is essential for expansion of DN thymocytes.** To directly assess the effects of *Txnrd1* deletion in hematopoietic precursors, we generated *Txnrd1^{fl/fl}*;*Cre-ERT2* mice that allow *Txnrd1* gene deletion at will by TAM injection. We evaluated the function of *Txnrd1* in early thymocyte development 2 weeks after TAM treatment. *Txnrd1* was completely deleted in T cells from Cre-expressing mice at the genomic DNA and mRNA levels upon TAM administration (Supplementary Fig. 2a,b). These mice had visibly smaller thymi (Fig. 2a) due to a massively reduced cellularity (Fig. 2b), which affected all thymocyte subsets, including DN and DP precursors, mature SP αβ T cells, γδ T cells, as well as iNKTs (Fig. 2c, d). Importantly, we excluded the possibility that reduction of thymocytes occurred due to Cre-mediated toxicity rather than deletion of *Txnrd1* in DN cells (Supplementary Fig. 2c). The massive reduction of DP precursors is consistent with the proliferation occurring during $CD44^-CD25^-$ (DN4) to DP phase (pre-α proliferation). In line with this, DN4 thymocytes lacking *Txnrd1* showed defective proliferation, measured as incorporation of the thymidine analog 5-Ethynyl-2'-deoxyuridine (EdU) (Fig. 2e). Analysis of known DN precursor subsets (i.e., $CD44^+CD25^-$ [DN1], $CD44^+ CD25^+$ [DN2], $CD44^-CD25^+$ [DN3] and DN4) showed that each was present in reduced numbers in TAM-treated *Txnrd1^{fl/fl}*;*Cre-ERT2* mice, although DN2–DN3 stage where pre-β proliferation occurs was affected the most, highlighting the significance of *Txnrd1* during proliferation (Fig. 2f).

To assess whether *Txnrd1* is cell-intrinsically required in T cells during development, we reconstituted irradiated WT mice ($CD45.1^+$) with an equal ratio of congenically marked BM cells from WT ($CD45.1^+CD45.2^+$) and TAM-treated *Txnrd1^{fl/fl}*;*Cre-ERT2* ($CD45.2^+$) or control *Txnrd1^{fl/fl}* ($CD45.2^+$) mice. In line with the previous observation, we found reduced contribution of *Txnrd1*-deficient cells during all stages of T cell development with the most prominent decrease from the DN3 stage (Supplementary Fig. 2d). As expected, abolished thymocyte development resulted in an almost complete absence of T cells in the periphery (Supplementary Fig. 2e). Taken all together, these results indicate that the Trx1 system has a critical function for proliferation during thymic development, in line with the expression levels of *Txn1*, *Txnrd1*, and *Txnip* in the different thymic populations (Fig. 1h and Supplementary Fig. 1h).

**Txnrd1 is critical for T-cell immune responses in vivo.** Given the requirement for expansion of T-cell precursors but not the homeostatic maintenance of naive αβ T cells, we next investigated the requirement of the gene during an in vivo immune response. To address this, we infected mice with lymphocytic choriomeningitis virus (LCMV)-WE strain. We found a drastic reduction in activated $CD62L^-CD4^+$ and $CD62L^-CD8^+$ T cells in spleen and blood of *Txnrd1^{fl/fl}*;*Cd4-Cre* compared to control mice (Fig. 3a and Supplementary Fig. 3a). Moreover, virus-specific $CD4^+$ and $CD8^+$ T cells were undetectable by $gp_{61-80}$ and $gp_{33-41}$ tetramer staining, suggesting a crucial role of *Txnrd1* in T-cell expansion (Fig. 3b and Supplementary Fig. 3b). To assess whether this impaired expansion in *Txnrd1*-deficient T cells arose from defective in vivo cycling or increased cell death, we next measured cell proliferation during the course of a viral infection by incorporation of EdU. A drastic reduction in EdU incorporation was observed in *Txnrd1^{fl/fl}*;*Cd4-Cre* T cells at all time points analyzed with the largest difference at the peak of expansion (7 days post infection; Fig. 3c and Supplementary Fig. 3c). Moreover, staining of 7-AAD and Annexin-V during infection showed comparable frequencies of dead/dying $CD4^+$ and $CD8^+$ T cells (Supplementary Fig. 3d,e). In line with the lack of virus-specific T cells, *Txnrd1^{fl/fl}*;*Cd4-Cre* mice failed to clear LCMV unlike control mice (Fig. 3d). Similarly, *Txnrd1* was also essential for Th1-mediated protection against the protozoan parasite *L. major*, as judged by increased footpad swelling (Fig. 3e), elevated parasite load (Fig. 3f), and impaired numbers of activated $CD4^+$ and $CD8^+$ T cells (Fig. 3g) in footpad and draining LNs of infected *Txnrd1^{fl/fl}*;*Cd4-Cre* mice. Taken all together, these data suggest a pivotal role of the Trx1 system in expanding T cells during in vivo immune responses.

**Induction of the Trx system upon T-cell activation.** To further elucidate mechanisms of defective T-cell expansion, we studied responses of naive T cells to αCD3/CD28 stimulation in vitro. TCR and IL-2R triggering are known to induce c-Myc which is essential for proliferation of T cells (as well as cancer and embryonic stem cells)[25,26] and metabolic reprogramming[21]. Expectedly, c-Myc expression was upregulated upon T-cell activation (Fig. 4a and Supplementary Fig. 4a). Concomitantly, both *Txn1* and *Txnrd1* but not Grx1 expression were upregulated (Fig. 4b, c), while the prominent expression of *Txnip* in naive T cells was abrogated within 4 h of T-cell activation (Fig. 4d), suggesting that the Trx system is the critical pathway controlling T-cell activation and expansion. *Txnrd1* deficiency did not alter *Txnip*, *Txn1*, and *c-Myc* expression indicating that TrxR1 acts

**Fig. 1** *Txnrd1* is required for thymic iNKT cell development. **a–c** T-cell populations in naive *Txnrd1^{fl/fl}*;*Cd4-Cre* and *Txnrd1^{fl/fl}* littermate control mice were analyzed by flow cytometry. Representative FACS plots (left) and quantification (right) are shown. **a** Thymic T-cell development was assessed by gating on $CD4^-CD8^-$ DN, $CD4^+CD8^+$ DP, $CD4^+TCRβ^+$ (CD4^+T), and $CD8^+ TCRβ^+$ (CD8^+T) thymocytes (*n* = 3–4). **b** Splenic $CD4^+$ and $CD8^+$ T cells (*n* = 3–4). **c** Expression of CD62L and CD44 on splenic $CD4^+$ (top) and $CD8^+$ (bottom) T cells (*n* = 4–5). **d, e** Lethally irradiated WT mice were reconstituted with a 1:1 mixture of WT and *Txnrd1^{fl/fl}*;*Cd4-Cre* bone marrow expressing the congenic markers CD45.1 and CD45.2, respectively. After reconstitution, the contribution of *Txnrd1^{fl/fl}*;*Cd4-Cre* cells to the indicated thymic and splenic T cell populations was assessed. Values were normalized to non−Cre expressing $CD45.2^+CD19^+$ B cells. Values below 1 indicate reduced contribution of *Txnrd1*-deleted T cells to the indicated population compared to WT cells (*n* = 4). **f** iNKT cells gated as $TCRβ^{int}CD1d-PBS57-tetramer^+$ in the thymi (top), livers (middle), and spleens (bottom) of naive *Txnrd1^{fl/fl}*;*Cd4-Cre* (or control) mice (*n* = 4–5). **g** Analysis of the indicated stages of MACS-enriched, $CD1d-PBS57-tetramer^+PLZF^+$ iNKT cells. Stage 0, 1, 2, and 3 iNKTs were gated as $CD24^{hi}CD44^{lo}NK1.1^-$, $CD24^{lo}CD44^{lo}NK1.1^-$, $CD24^{lo}CD44^{hi}NK1.1^-$, and $CD24^{lo}CD44^{hi}NK1.1^+$, respectively (*n* = 5–6). **h** Depicted are the expression levels of *Txn1* and *Txnip* determined by RT-PCR for FACS-sorted ETP ($lin^-CD44^{hi}c-Kit^{hi}CD25^-$), DN1-2 ($lin^-CD44^{hi}c-Kit^{hi}CD25^{int}$), DN2 ($lin^-CD44^{hi}c-Kit^{int/hi}CD25^{hi}$), DN2-3 ($lin^-CD44^{int}CD25^{hi}$), DN3A ($lin^-CD44^-CD28^-CD25^{hi}$), DN3B ($lin^-CD44^-CD28^-CD25^+$), DN3-4 ($lin^-CD44^-CD28^+CD25^{int}$), DN4 ($lin^-CD44^-CD28^+CD25^-$), ISP ($CD8^+CD24^+TCRβ^-$), DP blast ($CD4^+CD8^+FSC^{hi}$), DP rest ($CD4^+CD8^+FSC^{lo}$), $CD8^+$ and $CD4^+$ thymocyte populations from WT mice. Circular arrows indicate proliferating populations (*n* = 4). Bar graphs show mean + standard deviation. Data are representative of five (**a, b**), four (**c, f**), and two (**d, e, g, h**) independent experiments. Student's *t* test (two-tailed, unpaired) was used to compare *Txnrd1^{fl/fl}* and *Txnrd1^{fl/fl}*;*Cd4-Cre* groups (**a–c, f, g**): *$P \le 0.05$; **$P \le 0.01$; ***$P \le 0.001$; ****$P \le 0.0001$; ns not significant. One-sample *t* test with a hypothetical value of 1 was used in **d**: ****$P \le 0.0001$. One-way ANOVA adjusted by Tukey's multiple comparison test was used in **e**: *$P \le 0.0332$; ***$P \le 0.0002$; ****$P \le 0.0001$

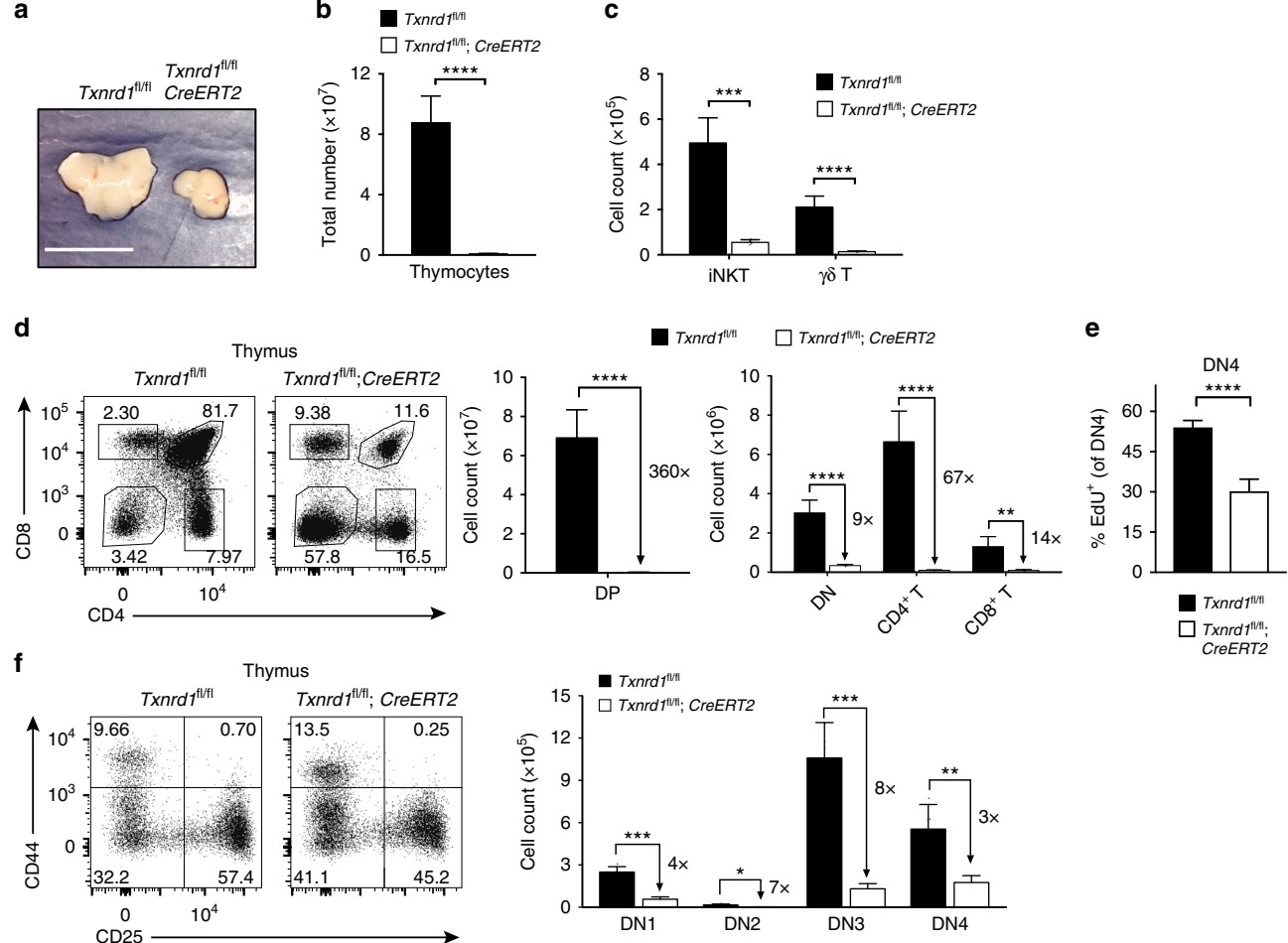

**Fig. 2** *Txnrd1* is critically required for thymic T-cell development. *Txnrd1*$^{fl/fl}$;*Cre-ERT2* mice (and *Txnrd1*$^{fl/fl}$ littermates as controls) were analyzed by flow cytometry 2 weeks after TAM-mediated *Txnrd1* deletion ($n = 4$–5). **a** Representative image of thymi (scale bar = 1 cm). **b** Total number of thymocytes. **c** Absolute cell counts of TCRβ$^{int}$CD1d-PBS57-tetramer$^+$ iNKT and TCRβ$^-$TCRγδ$^+$ (γδ T) cells in the thymus. **d** Representative FACS plots (left) and absolute numbers of CD4$^-$CD8$^-$ DN, CD4$^+$CD8$^+$ DP, CD4$^+$TCRβ$^+$ (CD4$^+$ T), and CD8$^+$TCRβ$^+$ (CD8$^+$ T) thymocytes (right). **e** Proliferation of DN4 thymocytes (CD25$^-$CD44$^-$) assessed by thymic EdU incorporation. **f** Representative FACS plots (left) and absolute cell numbers of CD25$^-$CD44$^+$ (DN1), CD25$^+$CD44$^+$ (DN2), CD25$^+$CD44$^-$ (DN3), and CD25$^-$CD44$^-$ (DN4; right). Bar graphs show mean + standard deviation. Numbers in the bar graphs next to the arrows indicate fold-changes between *Txnrd1*$^{fl/fl}$ and *Txnrd1*$^{fl/fl}$;*Cre-ERT2* groups (**d**, **f**). Data are representative of three (**a–d**, **f**) and two (**e**) independent experiments. Student's *t* test (two-tailed, unpaired) was used to compare *Txnrd1*$^{fl/fl}$ and *Txnrd1*$^{fl/fl}$;*Cre-ERT2* groups (**b–f**): *$P \leq 0.05$; **$P \leq 0.01$; ***$P \leq 0.001$; ****$P \leq 0.0001$; ns not significant

downstream and is not involved in T-cell activation (Fig. 4a–d). To assess a potential regulation of *Txnip* by c-Myc[27], we used JQ1, a transcriptional inhibitor of c-Myc. Indeed, JQ1 treatment of T-cell stimulation cultures dose-dependently inhibited c-Myc protein production and cell surface expression of CD71, a well-established c-Myc target gene (Fig. 4e). Notably, c-Myc blockade prevented *Txnip* downregulation induced by T-cell activation (Fig. 4f), indicating that TCR-triggering induction of c-Myc is responsible for Txnip repression.

In line with the in vivo proliferation defect, we observed lower numbers of T cells lacking *Txnrd1* compared to control T cells upon stimulation (Fig. 4g). This defect was confirmed by CFSE dilution analysis (Fig. 4h and Supplementary Fig. 4b). The application of the precursor cohort method[28] on this data allowed calculation of the mean division time ($t_{division}$) and the time to perform the first cell division ($t_{1°division}$) upon T-cell stimulation. We found that the mean division time in both CD4$^+$ and CD8$^+$ T cells lacking *Txnrd1* was ~30% longer. Moreover, completion of the first cell division was also delayed in the absence of *Txnrd1* (Fig. 4i and Supplementary Fig. 4c), suggesting that additional

defects in cell cycle entry might play a role. Consistently, stimulated *Txnrd1*-deficient T cells had a delayed upregulation of Ki-67, an indicator of cell cycle entry (Fig. 4j and Supplementary Fig. 4d). Finally, we also evaluated whether increased cell death in T cells lacking *Txnrd1* might contribute to impaired expansion. We found that viability of *Txnrd1*-deficient T cells was marginally affected (Supplementary Fig. 4e). Yet, the use of classical apoptosis, necroptosis, ferroptosis inhibitors, or anti-oxidants was insufficient to rescue this death (Supplementary Fig. 4f) and proliferation (not shown) indicating that absence of *Txnrd1* did not activate a pathway of controlled cell death.

The Trx system is a major player in cellular antioxidant responses[2]. To test whether the defective proliferation in *Txnrd1*-deficient T cells is a consequence of increased cellular reactive oxygen species (ROS) levels, we stimulated T cells from *Txnrd1*$^{fl/fl}$;*Cd4-Cre* and *Txnrd1*$^{fl/fl}$ mice in the presence of classical antioxidants. However, supplementation failed to restore proliferation of *Txnrd1*-deficient T cells (Fig. 4k). Moreover, generation of cellular ROS detected by CM-H$_2$DCFDA staining in stimulated T cells was comparable (Supplementary Fig. 4g).

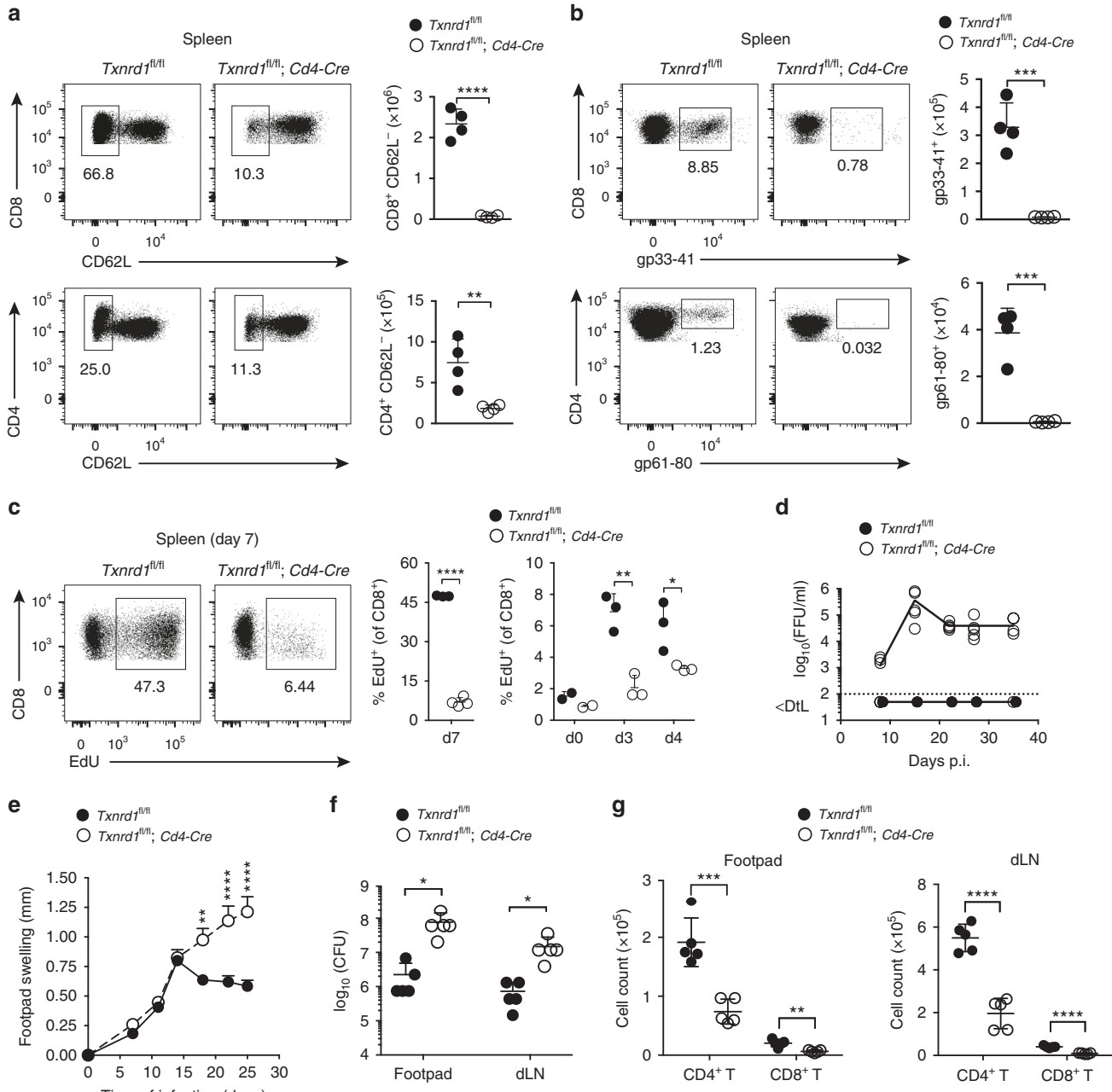

**Fig. 3** *Txnrd1*-deficient T cells fail to expand following viral and parasite infection. **a–c** *Txnrd1*fl/fl and *Txnrd1*fl/fl;*Cd4-Cre* mice were infected i.v. with 200 focus-forming units (FFU) LCMV WE. **a**, **b** On day 7 of infection, activation of CD4+ and CD8+ T cells was assessed by downregulation of CD62L (**a**), and virus-specific T cells were stained with gp33–41 and gp61–80 tetramers (**b**). **c** Proliferation of CD8+ T cells was assessed by EdU incorporation in spleens of LCMV-infected mice at the indicated days of infection. **d** LCMV titers over time were determined by focus assay of blood from mice infected with 500 FFU LCMV WE i.v. at indicated times after infection. Dotted line represents the detection limit (DtL); *n* = 5). **e–g** *Txnrd1*fl/fl and *Txnrd1*fl/fl;*Cd4-Cre* mice were inoculated with 2 millions *L. major* promastigotes into the right and left hind footpad. **e** Footpad swelling. **f** Parasite load in infected footpads and dLNs 25 days after infection. **g** Total number of activated (CD62L−CD44+) CD4+ and CD8+ T cells in footpads (left) and dLNs (right) 25 days after infection. Means + standard deviations and +standard error of the mean are shown for **a–c**, **f**, **g**, and for **e**, respectively. Data are representative of two independent experiments (**a–g**). Student's *t* test (two-tailed, unpaired) was used to compare *Txnrd1*fl/fl and *Txnrd1*fl/fl;*Cd4-Cre* groups (**a–c**, **f**, **g**): *P ≤ 0.05; **P ≤ 0.01; ***P ≤ 0.001; ****P ≤ 0.0001. Two-way ANOVA adjusted by Bonferroni's multiple comparison test was used to compare *Txnrd1*fl/fl and *Txnrd1*fl/fl;*Cd4-Cre* groups in **e**: **P ≤ 0.0021; ****P ≤ 0.0001

Since secreted Trx has been reported to induce IL-2R[29], we stimulated mixtures of T cells from congenically marked WT (CD45.1+) and *Txnrd1*fl/fl;*Cd4-Cre* (CD45.2+) mice at different ratios. Presence of *Txnrd1*-competent T cells did not rescue the defective expansion of *Txnrd1*-deficient T cells (Supplementary Fig. 4h), indicating a cell-intrinsic requirement of TrxR1.

Furthermore, CD25, CD44, and CD69 upregulation was comparable between *Txnrd1*-deficient and control T cells upon stimulation (Fig. 4l and Supplementary Fig. 4i), excluding impaired activation as a cause for defective expansion.

Overall, these results indicate that T-cell proliferation requires c-Myc-dependent unlocking of Trx1 from Txnip-mediated

inhibition and that *Txnrd1* deficiency delays cell cycle entry and slows down cell cycling independent of T-cell activation, cell death, and cellular ROS.

**Txnrd1 is required for nucleotide biosynthesis in T cells.** Growing evidence demonstrates that T-cell activation is coupled to drastic changes in cellular metabolism that regulate the

differentiation and function of activated T cells[20,30]. To explore whether the defective proliferation of *Txnrd1*-deficient T cells was the result of impaired metabolic reprogramming, we stimulated naive T cells in vitro and measured oxygen consumption rate (OCR), an indicator of oxidative phosphorylation (OXPHOS), and extracellular acidification rate (ECAR), a measure of aerobic glycolysis. We observed that *Txnrd1*-deficient T cells displayed substantially higher basal and maximal OCR compared to control

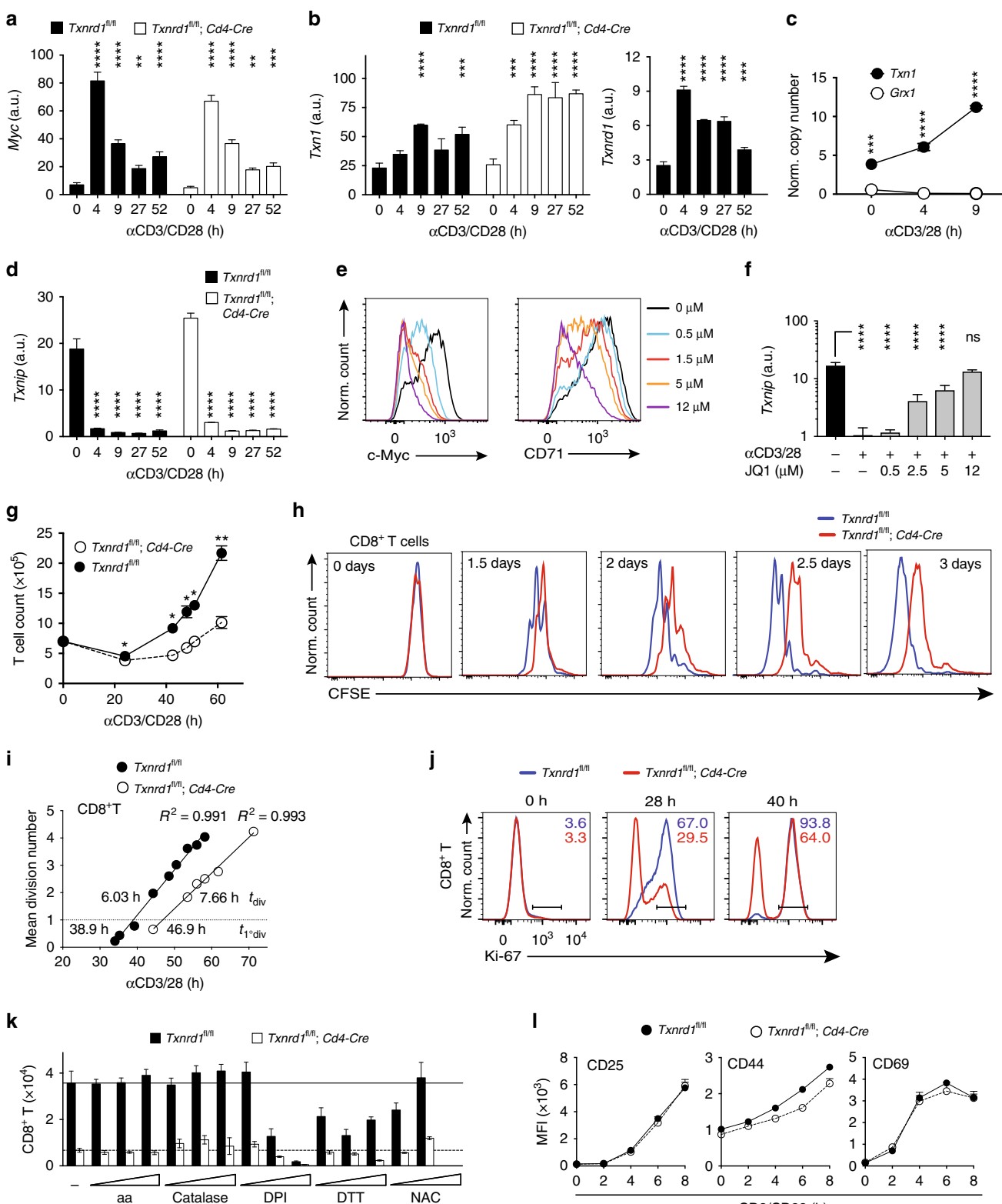

cells. Moreover, they also had higher baseline ECAR as well as maximal glycolytic capacity (Fig. 5a), indicating that *Txnrd1*-deficient T cells were metabolically more active compared to control cells. Phosphorylation of mTOR was comparable between *Txnrd1*-deficient and control T cells upon stimulation (Supplementary Fig. 5a). However, we observed increased phosphorylation of the metabolic stress sensor 5′AMP-activated protein kinase (AMPK) in the absence of the Trx system (Fig. 5b). We additionally measured decreased ATP-to-AMP ratio in *Txnrd1*-deficient T cells upon stimulation, indicating that AMPK activation is due to energy crisis (Fig. 5c). AMPK is well known to respond to nutrient availability and to reprogram cellular metabolism toward increased glycolysis, β-oxidation and mitochondrial function[31]. These data are therefore consistent with and explain the higher ECAR and OCR observed in *Txnrd1*-deficient T cells (Fig. 5a).

To perform a more comprehensive analysis of the metabolism in proliferating *Txnrd1*-deficient T cells and to understand the cause of the observed metabolic stress, we next utilized high-throughput, non-targeted, flow injection time-of-light metabolomics[32]. This method allows rapid and deep profiling of metabolites involved in central carbon metabolism and in peripheral pathways, such as urea cycle, amino acid metabolism, and nucleotide biosynthesis. We stimulated WT and *Txnrd1*-deficient T cells in vitro and extracted polar metabolites for metabolomics analysis. We identified a total of 989 distinct ion species and putatively annotated them to metabolites listed in the human metabolome database (HMDB) version 3.0[33]. Out of 989 metabolites, 101 were significantly changed between *Txnrd1*-deficient and control T cells upon stimulation (|Log2(fold-change)| > 0.5; adjusted $p$[Benjamini–Hochberg] < 0.01), but only eight in unstimulated conditions, indicating that the absence of the Trx system influences the metabolism of T cells mainly after stimulation (Supplementary Fig. 5b and Supplementary Data 1). To obtain a general overview of the metabolic differences, we performed a pathway enrichment analysis of the metabolites changing in *Txnrd1*-deficient versus control T cells both before and after stimulation. This revealed significantly increased levels of metabolites involved in purine and pyrimidine metabolisms in *Txnrd1*-deficient T cells (Fig. 5d). In addition, *Txnrd1*-deficient T cells also showed significant accumulation of metabolites involved in the aspartate and glutamate metabolism (Fig. 5d and Supplementary Fig. 5c). However, no significant enrichment for metabolic pathways that have been described to regulate immune cell function was observed, such as glycolysis, tricarboxylic acid (TCA) cycle, and fatty acid synthesis and oxidation (Fig. 5d and Supplementary Fig. 5d,e). Among the components of purine and pyrimidine biosynthetic pathways, we observed accumulation of

nucleotides containing ribose sugar, suggesting that the final reduction of RNA into DNA building blocks during DNA biosynthesis was defective (Fig. 5e). To confirm this, we performed a targeted metabolomics study measuring the cellular levels of the abundant purine ribonucleotides and 2′-deoxyribonucleotides. As expected, T-cell stimulation in the absence of the Trx system led to accumulation of ribonucleotides but reduced abundance of their 2′-deoxyribonucleotide forms (Fig. 5f). Taken together, these data imply that the Trx system in T cells regulates the reduction of RNA into DNA building blocks at the last step of DNA biosynthesis.

**Increased *Cdkn1a* expression in *Txnrd1*-deficient T cells.** To further investigate the cellular function of TrxR1 in T cells, we performed RNA sequencing (RNA-seq) of in vivo-activated T cells. Since virus-specific T cells failed to expand in the absence of *Txnrd1* during LCMV infection (Fig. 3b and Supplementary Fig. 3b), this model would not have allowed us to isolate enough cells for the experiment. To circumvent this issue, we injected SEB superantigen to obtain enough activated T cells for transcriptomics. After 2 days post SEB injection, Vβ8.1/8.2+CD8+ T cells from *Txnrd1*fl/fl;*Cd4-Cre* and *Txnrd1*fl/fl controls were FACS-sorted for analysis. Total CD8+ cells from PBS-treated animals served as a control. Similar to αCD3/CD28 T cell stimulation in vitro, superantigen-mediated T cell activation in vivo resulted in downregulation of *Txnip* and upregulation of *Txnrd1* and *Txn1* expression (Fig. 6a). Moreover, the frequency of Vβ8.1/8.2+ among CD8+ T cells was significantly reduced in *Txnrd1*fl/fl;*Cd4-Cre* upon SEB administration but not in the PBS-treated group, indicating defective expansion in the absence of *Txnrd1* (Fig. 6b and Supplementary Fig. 6).

Among the top differentially expressed gene sets (|GFOLD| > 1.5; false discovery rate [FDR] < 0.2) between SEB-stimulated *Txnrd1*-deficient and control T cell groups, the cell cycle inhibitor *Cdkn1a* (the gene encoding p21; FDR = 0.05) was significantly enriched in *Txnrd1*-deficient T cells (Fig. 6c). Analysis of the differentially expressed gene sets involved in cell-cycle progression and regulation pathways revealed that *Cdkn1a* was the only gene significantly enriched in *Txnrd1*-deficient T cells upon SEB stimulation (Fig. 6d). Conversely, levels of *Cdkn1a* expression in unstimulated T cells were comparable between *Txnrd1*-deficient and -sufficient T cells (Fig. 6d). Furthermore, *Cdkn1a* also appeared as the top single best predictor, when we used a linear regression model on all detected genes in SEB-treated groups as an additional statistical tool (Supplementary Table 1). Consistent with in vivo RNA-seq results, in vitro T cell stimulation with αCD3/CD28 and real-time quantitative PCR (RT-PCR)

---

**Fig. 4** *Txnrd1*-deficient T cells have a proliferation defect independent of ROS. Magnetically sorted T cells from spleen and peripheral LNs of *Txnrd1*fl/fl and *Txnrd1*fl/fl;*Cd4-Cre* mice were stimulated in αCD3/CD28-coated plates. **a, b** Shown are the expression levels of *c-Myc* (**a**), *Txn1* and *Txnrd1* (**b**) determined by RT-PCR in CD4+ T cells ($n = 3$–4). **c** Normalized copy numbers of *Txn1* and *Grx1* determined by RT-PCR. The obtained Ct values were converted to DNA concentration using a standard curve ($n = 3$–4). **d** Expression level of *Txnip* determined by RT-PCR in CD4+ T cells ($n = 3$–4). **e** c-Myc (left) and CD71 (right) protein expression levels measured by flow cytometry after incubation of stimulating CD4+ T cells with the indicated concentrations of the c-Myc inhibitor (+)-JQ1 for 10 h. **f** *Txnip* expression levels in CD4+ T cells stimulated in the presence of the indicated concentrations of (+)-JQ1 ($n = 3$). **g** T cell counts at indicated times ($n = 2$). **h** CFSE dilution of CD8+ T cells measured by flow cytometry at the indicated times of stimulation. **i** CFSE-dilution profiles were used to calculate mean division time ($t_{division}$; reciprocal of the slope) and time to first division ($t_{1°division}$; intersection with mean division number $y = 1$) with the precursor cohort method. **j** CD8+ T cells leaving $G_0$ phase and entering cell cycle were detected by staining for Ki-67 at the indicated times of αCD3/CD28 stimulation. Numbers in the plots indicate average percentage of Ki-67+ cells from triplicate samples. **k** CD8+ T cells were treated with the antioxidants ascorbic acid (0.8, 4, 20 μM), Catalase-polyethylene glycol (catalase; 278, 1136, 4545 U/ml), Diphenyleneiodonium chloride (DPI; 62.5, 250, 1000 nM), DL-Dithiothreitol (DTT; 111, 333, 1000 μM) and N-acetyl-L-cysteine (NAC; 2, 8, 32 mM) during stimulation and counted after 3 days ($n = 4$). **l** Expression of indicated activation markers in CD8+ T cells measured by flow cytometry during the first 8 h of stimulation ($n = 3$). Dots and bar graphs are mean ± standard deviation. Data are representative of two (**a–g, k, l**), five (**h**) and three (**i, j**) independent experiments. One-way ANOVA followed by Dunnett's correction was used in **a, b, d**: **$P \le 0.0021$; ****$P \le 0.0001$. Student's $t$ test (two-tailed, unpaired) was used in **c, g**: *$P \le 0.05$; **$P \le 0.01$; ***$P \le 0.001$; ****$P \le 0.0001$. One-way ANOVA adjusted by Tukey's multiple comparison test was used in **f**: ****$P \le 0.0001$; ns not significant

confirmed higher expression levels of *Cdkn1a* in the absence of *Txnrd1* (Fig. 6e). Since p21 is known to play a critical role in the DNA damage response by inducing cell cycle arrest[34], these data are consistent with the slower proliferative capacity and indicate increased DNA damage response in *Txnrd1*-deficient T cells.

In addition to *Cdkn1a*, the expression levels of the Nrf2 target genes *Nqo1* and *Srxn1* were also increased in *Txnrd1*-deficient T cells, as suggested by RNAseq (Fig. 6c) and confirmed by RT-PCR (Supplementary Fig. 7a,b), indicating oxidative stress-induced Nrf2 activation as the response to maintain redox balance.

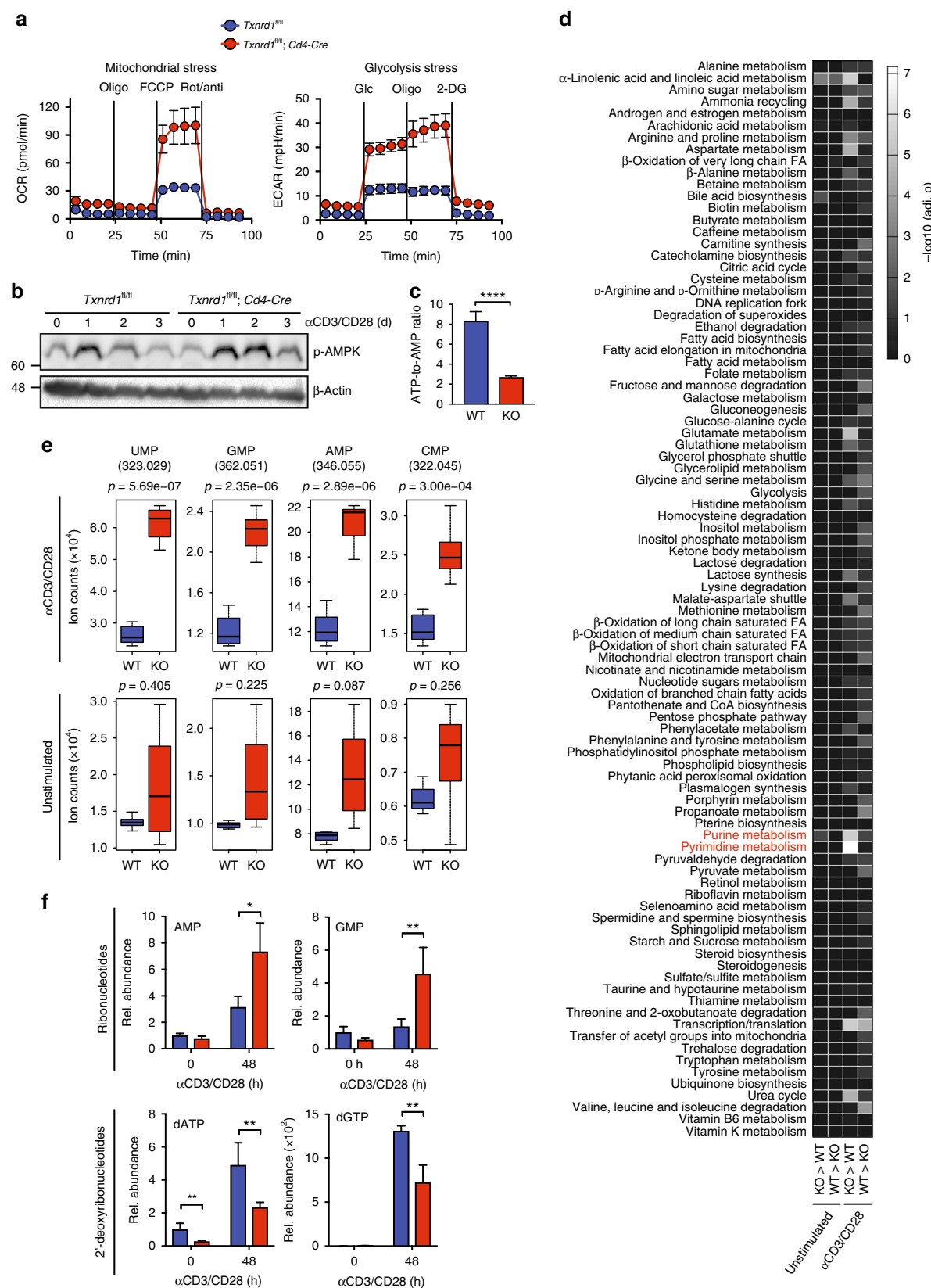

**Increased replication stress in *Txnrd1*-deficient T cells**. It is well established that low levels of DNA building blocks contribute to replication fork stalling and consequently activation of the DNA damage response. This process involves phosphorylation of the histone family member H2AX, leading to accumulation of its phosphorylated form, γH2AX[35,36]. Consistently, stimulation of normal T cells in the presence of hydroxyurea (HU) dose-dependently increased H2AX phosphorylation (Fig. 7a). HU specifically blocks the catalytic activity of RNR, the enzyme that reduces RNA into DNA building blocks in the last step of nucleotide biosynthesis[37,38]. Indeed, T cells with higher replication stress accumulated in the S-phase of the cell cycle (i.e., intermediate levels of DNA in the FACS plot) during which DNA building blocks are most required (Fig. 7a).

We hypothesized that defective nucleotide biosynthesis in *Txnrd1*-deficient T cells (Fig. 5) would cause increased replication stress and phosphorylation of H2AX, leading to enhanced expression of p21 (as shown in Fig. 6), similar to HU-treatment of WT T cells. To explore this possibility, we first assessed the role of *Txnrd1* in the progression of in vitro activated T cells through the different cell cycle stages. Notably, T cells lacking *Txnrd1* accumulated in the S-phase upon stimulation and were conversely reduced in the G0/G1- and G2/M-phases (Fig. 7b). Furthermore, *Txnrd1*-deficient T cells accumulating in the S-phase had higher levels of γH2AX (Fig. 7c) and increased expression of the DNA damage sensors *Ddb2* and *Xpc* compared to control cells (Supplementary Fig. 8a). Comparable γH2AX levels in T cells from *Txnrd1*^fl/fl^ and *Cd4-Cre* mice excluded the possibility that Cre-mediated toxicity/DNA damage rather than *Txnrd1* deletion was responsible for increased H2AX phosphorylation (Supplementary Fig. 8b). In line with these in vitro results, in vivo activation using α-CD3 administration similarly led to increased phosphorylation of H2AX among proliferating cells (Ki-67+). As a control, unstimulated T cells did not accumulate γH2AX irrespectively of the presence or absence of *Txnrd1* (Fig. 7d). To address whether the increased γH2AX level in the absence of the Trx system was due to stalled replication forks, we measured phosphorylation of ATR, the sensor that responds to this type of DNA damage[39]. We observed higher ATR phosphorylation in *Txnrd1*-deficient T cells (Supplementary Fig. 8c), in line with the lower amount of 2′-deoxyribonucleotides and the increased phosphorylation of H2AX. Taken together, these data show increased replication stress in the absence of the Trx system, thereby mimicking the HU-mediated phenotype caused by reduction of DNA building block pools.

Our results are consistent with previous reports indicating the Trx and the GSH systems as electron donors for RNR in bacteria[40,41]. Therefore, we next investigated to which extent the two cellular antioxidant systems are involved in nucleotide biosynthesis in T cells. We stimulated *Txnrd1*-deficient and control T cells in the presence of L-buthionine-sulfoximine (BSO), which blocks GSH synthesis by inhibiting glutamate-cysteine ligase, thereby reducing the cellular GSH levels[42]. We observed that BSO treatment further reduced the expansion capacity of *Txnrd1*-deficient T cells in a concentration-dependent manner. Conversely, control T cells were resistant to BSO treatment and were able to expand at BSO concentrations that completely blocked proliferation of cells lacking *Txnrd1* (Fig. 7e). BSO-treatment further increased the already higher γH2AX levels in *Txnrd1*-deficient T cells in a concentration-dependent manner, leading to cell-cycle arrest in the S-phase (Supplementary Fig. 8d). This demonstrated that blocking GSH in addition to the Trx system led to even higher replication stress. Notably, BSO treatment on control T cells did not affect levels of γH2AX (Supplementary Fig. 8d). Taken all together, these data demonstrate that the Trx system is the major electron donor for RNR during nucleotide biosynthesis in stimulated T cells. In contrast, the GSH system is dispensable and only inefficiently compensates for the lack of the Trx system in T cells.

## Discussion

T-cell activation and proliferation are coupled to drastic changes in cellular metabolism to support their increased bioenergetic and biosynthetic demands. Although less efficient than OXPHOS at producing ATP, activated T cells switch to aerobic glycolysis to generate metabolic intermediates that are crucial for cell growth and proliferation[20,30,43,44]. In particular, glucose-6-phosphate generated in the first step of glycolysis can be metabolized in the pentose phosphate pathway to generate ribose, thereby promoting nucleotide biosynthesis[20,21]. RNR catalyzes the rate-limiting step of nucleotide biosynthesis by reducing ribonucleotides into the corresponding 2′-deoxyribonucleotides[10]. Both the Trx and Grx systems are known electron donors for this reaction[40,41]. Previous studies showed that proliferation and development of embryonic fibroblasts, cardiomyocytes, and hepatocytes was unaffected in the absence of *Txnrd1*[45–47] and redundantly covered by the Grx system[48]. Through a mass-spectrum-based metabolomics approach, we found impaired pyrimidine and purine biosynthesis after T-cell stimulation in the absence of *Txnrd1* (Fig. 5), consistent with the Trx1 system at providing reducing equivalents to RNR. Glutamine and aspartate are crucial nitrogen donors for synthesis of purines and pyrimidines and can potentially accumulate in case RNR-catalyzed reaction is impaired[49,50]. Interestingly, metabolomics revealed also the accumulation of metabolites in the glutamate and aspartate metabolism, further indicating defective nucleotide biosynthesis in the absence of the Trx1 system. Seahorse-based measurements showed that *Txnrd1*-deficient T cells were metabolically more active, in line with a recent publication indicating that mouse embryonic fibroblasts lacking *Txnrd1* displayed increased metabolic flux, glycogen storage, lipogenesis, and adipogenesis[51]. Moreover, the higher metabolic activity in *Txnrd1*-

**Fig. 5** *Txnrd1*-deficient T cells have defective nucleotide biosynthetic pathway. Magnetically sorted T cells from spleen and peripheral LNs of *Txnrd1*^fl/fl^ (WT) and *Txnrd1*^fl/fl^;*Cd4-Cre* mice (KO) were stimulated in αCD3/CD28-coated plates for 2 days in vitro. **a** Oxygen consumption rate (OCR; *left*) and extracellular acidification rate (ECAR; *right*) were measured with a Seahorse Extracellular Flux XF-96 analyzer under basal conditions and following addition of the indicated metabolic inhibitors ($n = 6$; data are mean + standard deviation). **b** Western blot of p-AMPK expression with β-actin as a loading control in T cells. **c–e** Metabolites were analyzed by non-targeted metabolomics (four replicates were examined, each analyzed twice). **c** Depicted is the ATP-to-AMP ratio. **d** Metabolic pathway enrichment analysis comparing WT and KO T cells both before and after stimulation. *KO > WT* and *WT > KO* indicate higher metabolite levels in KO and WT T cells, respectively. **e** Box plots displaying cellular ribonucleotide levels in WT and KO T cells. Numbers in brackets represent the mass-to-charge ratio of the detected ions. **f** Ribonucleotide and 2′-deoxyribonucleotide metabolites were analyzed by targeted metabolomics at 0 and 48 h after stimulation ($n = 4$). Bar graphs are mean + standard deviation (**c**, **f**). The boxes represent the 25th to 75th percentiles, the horizontal lines within the boxes indicate the medians, and the whiskers are drawn to indicate the 1.5 × Inter Quartile Range (3rd quartile–1st quartile; **e**). Data are representative of two independent experiments (**a–f**). Student's *t* test (two-tailed, unpaired) was used in **c**, **f**: *$P \leq 0.05$; **$P \leq 0.01$; ****$P \leq 0.0001$. *P*-values were adjusted using Benjamini–Hochberg procedure (**d**). Significance of the box plot data was assessed using a Student's *t* test (two-tailed, unpaired) corrected by the Benjamini–Hochberg multiple testing (**e**)

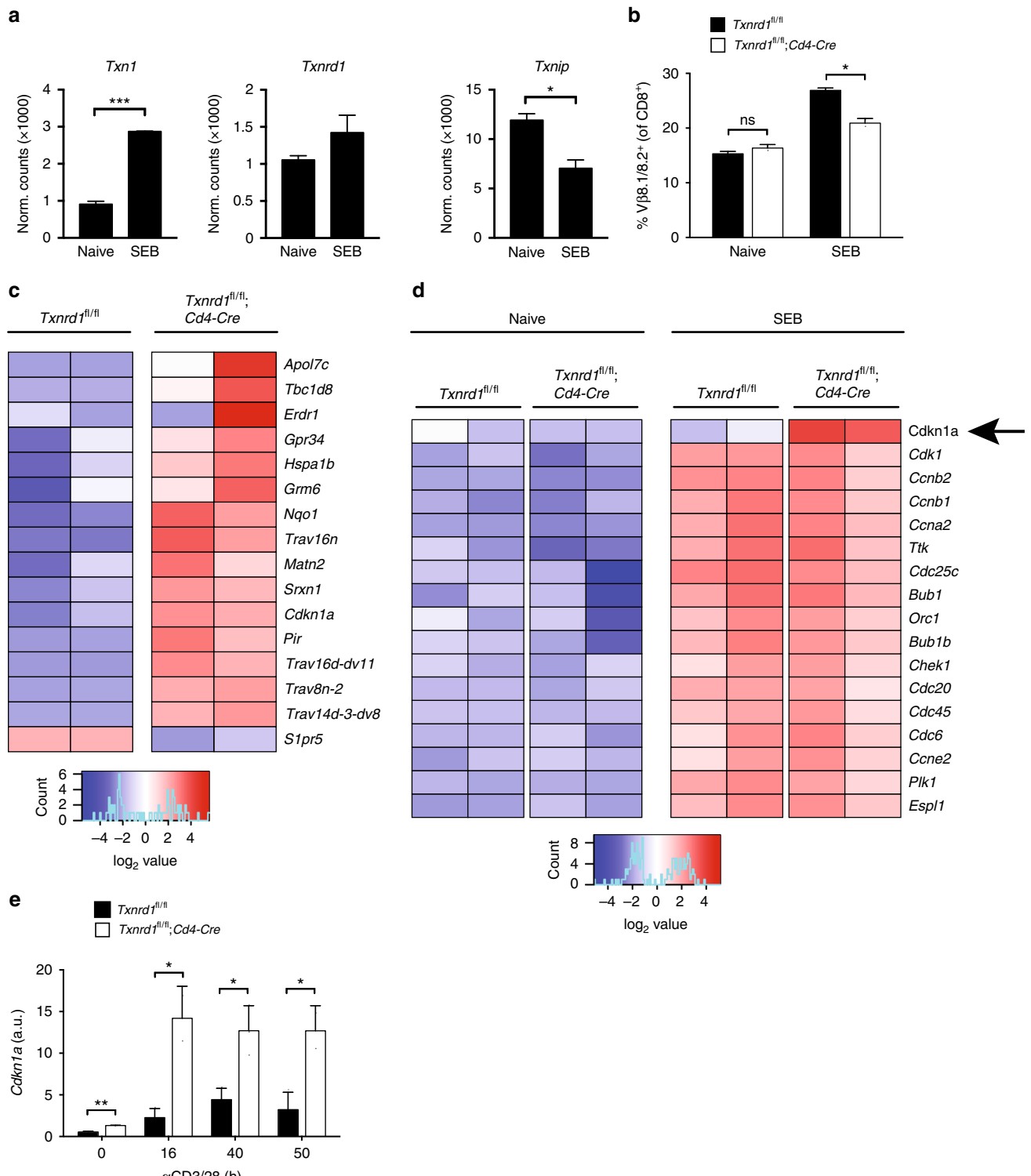

**Fig. 6** *Txnrd1* deficiency increases expression of the cell cycle inhibitor *Cdkn1a*. **a–d** *Txnrd1*<sup>fl/fl</sup> and *Txnrd1*<sup>fl/fl</sup>;*Cd4-Cre* mice were treated i.p. with 100 μg of Staphylococcal enterotoxin B (SEB) or PBS. Vβ8.1/Vβ8.2⁺CD8⁺ T cells were FACS sorted at day 2, and gene expression profiles determined by RNA-seq (two biological replicates per group). **a** Expression of *Txn1*, *Txnrd1*, and *Txnip* in Vβ8.1/V8.2⁺ CD8⁺ T cells from WT mice after PBS (naive) and SEB treatment. **b** Percentages of Vβ8.1/Vβ8.2⁺ cells among CD8⁺ T cell population are shown. **c** Heat map showing expression changes between SEB-stimulated *Txnrd1*-deficient and control T cell groups (|GFOLD| > 1.5; FDR < 0.2). **d** Heat map indicating the top differentially expressed genes involved in the cell cycle. Only genes with a |GFOLD| larger than 1.5 between naive and stimulated groups are displayed. **e** Magnetically sorted T cells from spleen and peripheral LNs of naive *Txnrd1*<sup>fl/fl</sup> and *Txnrd1*<sup>fl/fl</sup>;*Cd4-Cre* mice were stimulated in αCD3/CD28-coated plates. *Cdkn1a* expression was assessed using RT-PCR at the indicated times ($n = 2$–3). Bar graphs show mean + standard deviation (**a**, **b**, **e**). Data are representative of one (**a**, **c**, **d**) and two (**b**, **e**) independent experiments. Student's *t* test (two-tailed, unpaired) was used to compare *Txnrd1*<sup>fl/fl</sup> and *Txnrd1*<sup>fl/fl</sup>;*Cd4-Cre* groups (**a**, **b**, **e**): *$P ≤ 0.05$; **$P ≤ 0.01$; ***$P ≤ 0.001$; ns not significant

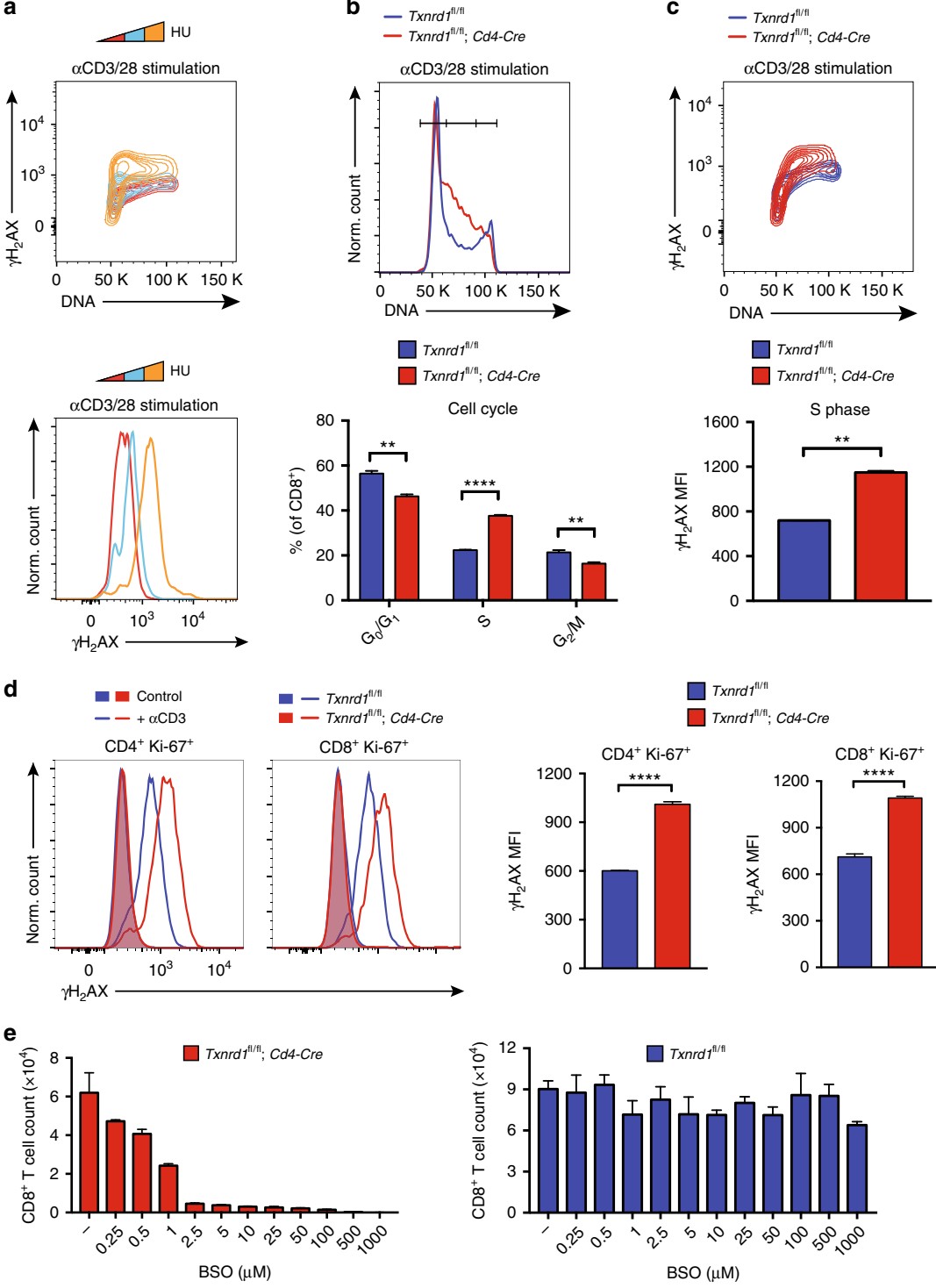

**Fig. 7** Activated *Txnrd1*-deficient T cells have an increased replication stress. **a–c** Magnetically sorted T cells from spleen and peripheral LNs of *Txnrd1*fl/fl and *Txnrd1*fl/fl;*Cd4-Cre* mice were stimulated in αCD3/CD28-coated plates in vitro. Levels of the DNA damage marker γH$_2$AX and DNA content were assessed by flow cytometry. **a** CD8[+] WT T cells were αCD3/28-stimulated in the presence of hydroxyurea (HU; 0, 10, 100 μM) for 30 h before analysis. **b** Cell cycle analysis of CD8[+] T cells with representative FACS plot with gating strategy (top; pre-gated on CD8[+]) and quantification (bottom) showing percentage of cells in the indicated cell cycle stages (*n* = 3). **c** Representative FACS plot of CD8[+] T cells (top) and MFI of γH$_2$AX in S phase cells after 30 h αCD3/28 stimulation (bottom; *n* = 3). **d** *Txnrd1*fl/fl and *Txnrd1*fl/fl;*Cd4-Cre* mice were injected i.v. with αCD3 antibody (10 μg), and γH$_2$AX were stained in splenic T cells 24 h after injection. Representative FACS plots (left) show γH$_2$AX expression in CD4[+]Ki-67[+] or CD8[+]Ki-67[+] T cells (+αCD3) compared to total CD4[+] or CD8[+] T cells from PBS-treated mice (Control). Bar graphs on the right quantify γH$_2$AX in CD4[+]Ki-67[+] or CD8[+]Ki-67[+] T cells from αCD3-treated mice (*n* = 3). **e** CD8[+] T cells were stimulated in the presence of increasing concentrations of BSO. After 3 days of αCD3/28 stimulation, live cells were counted (*n* = 4). Bar graphs show mean + standard deviation (**b–e**). Data are representative of two (**a**, **d**, **e**) and four (**b**, **c**) independent experiments. Student's *t* test (two-tailed, unpaired) was used to compare *Txnrd1*fl/fl and *Txnrd1*fl/fl;*Cd4-Cre* groups (**b–d**): **$P \leq 0.01$; ****$P \leq 0.0001$

deficient T cells is consistent with the observed increased phosphorylation of the metabolic stress sensor AMPK, thereby boosting glycolytic flux, β-oxidation, and mitochondrial function[31]. Increased glutamine levels in the absence of *Txnrd1* may also fuel the TCA cycle and potentially contribute to the increased OCR. Consistent with impaired nucleotide biosynthesis, we detected increased phosphorylation of H2AX in the S-phase of the cell cycle in the absence of *Txnrd1* after stimulation (Fig. 7). During normal cell-cycle progression, γH2AX decorates single-stranded DNA at the replication forks. The higher levels of γH2AX in the absence of *Txnrd1*, which was mimicked by direct inhibition of RNR by HU, is consistent with replication fork stalling events due to reduced availability of DNA building blocks[35,36]. Moreover, expression of the cyclin-dependent kinase inhibitor p21, known to participate in DNA damage response downstream of γH2AX[34], was upregulated (Fig. 6). Thus, in the absence of *Txnrd1*, cellular pools of DNA building blocks are depleted, thereby increasing replication stress, which in turn promotes p21-dependent inhibition of cell cycle progression. Taken together, our results demonstrate that the Trx system is the major electron donor for RNR during nucleotide biosynthesis in rapidly proliferating T cells during development and immune responses. Indeed, blockade of Grx system by targeting gamma-glutamylcysteine synthetase with BSO did not affect proliferation of WT T cells, while it abrogated the remaining proliferative capacity observed in activated *Txnrd1*-deficient T cells in vitro. Notably, this Grx-dependent partial capacity of *Txnrd1*-deficient T cells to proliferate in vitro was not observed upon LCMV infection in vivo, where expansion of anti-viral T cells was completely abrogated in the absence of *Txnrd1*. By contrast, Grx1 was poorly expressed in peripheral T cells and not upregulated upon T-cell activation, thereby providing an explanation why the Grx system cannot fuel DNA synthesis required for T-cell expansion.

Txnip plays a central role in the regulation of glucose metabolism and glycolysis by repressing cellular glucose uptake[14–17]. Naive T cells utilizing low glucose and OXPHOS for energy generation[20] showed high *Txnip* expression, while T-cell stimulation resulted in rapid *Txnip* downregulation, in line with activated T cells requiring glucose uptake during metabolic reprogramming[30]. We found that the transcription factor c-Myc, which controls the metabolic reprogramming occurring in activated T cells[21], links T-cell activation with downregulation of *Txnip*, in line with other reports demonstrating the potential of c-Myc to directly repress Txnip in tumor cells[27]. Increased glycolysis due to Txnip downregulation may dynamically deplete glucose metabolites that are sensed by chREBP–Mlx or MondoA–Mlx complexes[16,17] preventing *Txnip* gene expression and thereby generating a positive feedback loop further promoting glucose uptake and augmenting the flux of glycolysis[52]. In addition to facilitate glucose import, diminishment of *Txnip* releases Trx1 from inhibition. Concomitant with *Txnip* downregulation, T-cell activation resulted in *Txn1* upregulation. Trx1 is essential for the activation of the low energy sensor AMPK during metabolic stress by suppressing its oxidation[53]. Activated AMPK phosphorylates Txnip resulting in its degradation, which, in turn, increases Glut1 function and glucose import[15]. This mechanism may sustain proliferation in activated T cells. Consistently, T cells lacking AMPKα1 show strikingly reduced expansion upon viral and bacterial infection in vivo[54], similar to *Txnrd1*-deficient T cells.

The Trx system has also been proposed to inhibit apoptosis by binding and inactivating ASK1[9,55], and to maintain survival of DP thymocytes[56]. However, our data revealed no relevant increase of cell death in *Txnrd1*-deficient T cells upon activation in vitro and in vivo, and the numbers and viability of DP cells were unaffected in thymi of *Txnrd1fl/fl;Cd4-Cre* mice. Furthermore, as the Trx system is one of the main antioxidant pathways in mammalian cells, we originally speculated that increased ROS levels and consequent aberrant cellular functions were the major cause of the impaired proliferation of *Txnrd1*-deficient T cells. Yet, supplementation of antioxidants was found ineffective and ROS levels were comparable in WT and *Txnrd1*-deficient T cells, indicating that defective proliferation is independent of cellular ROS. It is feasible that the GSH system acts as the main cellular antioxidant[9] and protects T cells from ROS-mediated cell death. Indeed, crucial roles of the GSH system in the regulation of the antioxidant response and in promoting survival of activated T cells have been shown recently. GSH buffers ROS in order to promote c-myc-dependent metabolic reprogramming[57]. Moreover, deficiency of Glutathione peroxidase 4 results in fatal accumulation of membrane lipid peroxides and death by ferroptosis upon T-cell activation[58]. Thus, our results point to the essential requirement of the Trx system for donating electrons to RNR during rapid T-cell proliferation, while the GSH system is indispensable as antioxidant and for survival of activated T cells.

Trx and TrxR upregulation has been reported in several types of cancers, including lymphomas and aggressive T-cell acute lymphoblastic leukemia (T-ALL), and it was associated with a poor clinical outcome[59,60]. The Trx system has been suggested to promote development and propagation of cancer by various mechanisms including inhibition of apoptosis, promotion of cell growth, and sustainment of angiogenesis[59–61]. Several chemotherapeutic drugs that target TrxR are in the clinics and are considered to mediate antitumor activity primarily through induction of oxidative stress and apoptosis[62,63]. However, abrogation of T-cell expansion in the absence of *Txnrd1* was not associated with a significant increase in ROS production and cell death possibly indicating off-target effects of the inhibitors. Indeed, currently, there is no clinical drug that targets TrxR1 specifically. Our data suggest that such inhibitors may be efficacious and sufficient in treatment of lymphoma and leukemia by direct inhibition of DNA synthesis.

In conclusion, we propose a model where c-Myc-dependent Txnip downregulation by T-cell activation is critical for release of active Trx1 that in turn provides electrons for the synthesis of 2′-deoxyribonucleotides independently of the GSH–Grx system.

## Methods

**Mice.** *Txnrd1fl/fl* mice[46] were provided by M. Conrad (Helmholtz Zentrum, Munich, Germany) and were backcrossed for more than eight generations to C57BL/6. To obtain *Txnrd1fl/fl;Cd4-Cre* and *Txnrd1fl/fl;Cre-ERT2* mice, *Txnrd1fl/fl* mice were crossed with *Cd4-Cre*[64] and *Cre-ERT2* mice[65], respectively. B6 *Ptprca* (CD45.1) animals were purchased from The Jackson Laboratory (Bar Harbor, Maine, USA). About 6–12-week-old age- and sex-matched mice (either male or female) were used for the experiments. For deletion of *Txnrd1* in *Txnrd1fl/fl;Cre-ERT2* mice, mice were treated with 2 mg TAM (Sigma-Aldrich) intraperitoneally (i.p.) on two consecutive days. For EdU labeling, mice were injected i.p. with 0.5 mg EdU (Life Technologies) and analyzed 4 or 16 h later for thymic expansion or T-cell proliferation during LCMV infection, respectively. Mice were housed in individually ventilated cages under specific pathogen free conditions at ETH Phenomics Center (EPIC) (Zurich, Switzerland). All animal experiments were approved by the local animal ethics committee (Kantonales Veterinärsamt Zürich, licenses 167/2011, ZH270/2014, 113/2012, ZH135/15, and ZH161/15), and performed according to local guidelines (TschV, Zurich) and the Swiss animal protection law (TschG).

**Bone marrow chimeras.** CD45.1+ or CD45.1+ CD45.2+ recipients were irradiated twice with 4.75 Gy with a 4-h break in a RS 2000 (Rad Source Technologies Inc., Alpharetta, USA). The following day, mice were reconstituted by intravenous (i.v.) injection of bone marrow cells from hind legs of donor mice, treated with antibiotics (0.024% Borgal, MSD Animal Health, in the drinking water) for 6 weeks, and used for experiments 8 or more weeks after reconstitution. *Txnrd1fl/fl;Cre-ERT2* (and *Txnrd1fl/fl* as control) donor mice were pre-treated twice with 2 mg TAM i.p. to delete *Txnrd1* gene before BM transfer.

**LCMV and *Leishmania major* infection**. LCMV WE was originally provided by Rolf Zinkernagel (University of Zurich, Zurich, Switzerland) and propagated in L929 cells with virus aliquots stored at −80 °C. Mice were i.v. infected with 200 or 500 FFU LCMV WE. Blood was diluted 1:5 in MEM medium (Life Technologies) + 2% FCS (Gibco) was stored at −80 °C until LCMV titers were determined in MC57 cells as previously described[66].

*L. major* parasite (MHOM/IL/81/FEBNI) were grown in Schneider's *Drosophila* medium (Invitrogen) with 20% FBS and 100 U/ml penicillin 6-potassium and 100 μg/ml streptomycin sulfate. Mice were injected with 2 millions promastigote parasites into the left and right hind footpad, and they were analyzed 25 days after infection. Footpad swelling was monitored every 3–4 days. To quantify parasite load in footpads and draining LNs, single-cell suspensions were plated on 96-well plates in duplicates of threefold serial dilutions and were cultured at 30 °C for 7 days.

**In vivo stimulation**. In total, 100 μg Staphylococcal enterotoxin B (Sigma-Aldrich) was injected i.p. per mouse and blood/spleen were analyzed at indicated days. For DNA damage analysis in vivo, 10 μg of anti-CD3 (145-2C11; home-made) was injected i.v., and splenocytes were isolated 24 h later for staining.

**Flow cytometry**. The LCMV glycoprotein peptides gp$_{33−41}$ (KAVYNFATM), gp$_{61−80}$ (GLNGPDIYKGVYQFKSVEFD), and CD1d-PBS57 tetramer were kindly provided by the National Institutes of Health (Bethesda, MD). Fixable viability dye eFluor® 780 (eBioscience) and LIVE/DEAD® fixable yellow dead cell stain (Life Technologies) were stained in pure PBS before antibody staining. For EdU proliferation experiments, cells from EdU-injected mice were labeled before staining with antibodies with the Click-iT Plus EdU Alexa Fluor® 488 Flow Cytometry Assay Kit (Life Technologies) according to manufacture's instructions. Antibodies for extracellular stains were incubated with cells for 15 min in FACS buffer (PBS + 2% FCS). For c-Myc, γH2AX, PLZF, pATR, and Ki-67 intracellular stainings, cells were fixed and permeabilized with Foxp3 Staining Buffer Set (eBioscience) according to manufacture's instructions and subsequently stained for 30 min. The anti-pATR and the anti-c-Myc antibodies were detected with an anti-rabbit secondary antibody labeled with the FITC fluorophore. To analyze cell death, cells were stained with Annexin-V-APC (BD Bioscience) and 7-AAD (eBioscience) in Annexin-V binding buffer. For cell cycle analysis, DNA was stained with Fxcycle violet stain (Thermo Scientific) according to manufacture's instructions. Total cellular ROS was quantified by adding 1 μM CM-H$_2$DCFDA (Life Technologies) to cells and measuring fluorescein signal after 45 min recovery in supplemented medium. Cells were acquired on FACSCanto II or LSRFortessa (BD Bioscience), or sorted on FACSAria III (BD Bioscience). Data were analyzed in FlowJo software (Tree Star). A complete list of all antibodies and staining reagents used in this study can be found in Supplementary Table 2. All the gating strategies used for flow cytometry plots are shown in Supplementary Figs 9–13.

**Magnetic cell sorting**. For analysis of the different iNKT developmental stages, single thymi were MACS-enriched with APC-labeled CD1d-PBS57-tetramer and anti-APC microbeads (MACS, Miltenyi Biotec). For T-cell stimulation experiments in vitro, T cells were enriched from spleen and lymph node (inguinal, axillary, brachial) by positive selection using a MACS system with microbeads conjugated to monoclonal anti-mouse CD4 (L3T4), CD8a (Ly-2), or CD90.2 (Miltenyi Biotec) following the manufacturer's instructions. Briefly, 100 millions cells/ml were stained with microbeads (1:10 dilution) for 15 min at 4 °C. After washing, 100 millions cells were resuspended in 500 μl of buffer, and the cell suspension was applied to LS columns and positively sorted in the magnetic field of a MACS separator.

**In vitro T-cell stimulation**. MACS-sorted T cells (10$^5$/well) were seeded in IMDM + GlutaMAX, 10 % FCS, 100 U/mL penicillin, 100 μg/mL streptomycin, 50 μM β-mercaptoethanol (all Gibco) in 96 well plates pre-coated with anti-CD3 (4 μg/ml; 145-2C11; home-made) and anti-CD28 (2 μg/ml; 37.51; home-made) for stimulation. For time-course experiments, cells were fixed with 4% Formaldehyde in PBS (Sigma-Aldrich). For CFSE-labeling, 10$^7$/ml PBS-washed cells were incubated for 10 min at 37 °C with 5 μM CFSE (5-(and-6)-Carboxyfluorescein Diacetate, Succinimidyl Ester; life technologies) in PBS + 0.1% BSA (bovine serum albumin; Sigma-Aldrich). Analysis of proliferation kinetics of CFSE-labeled cells was performed as described elsewhere (precursor cohort method[28]). Inhibitors and compounds were added at the beginning of culture. The following substances were used (purchased from Sigma-Aldrich unless indicated otherwise): L-ascorbic acid, L-Buthionine-sulfoximine, Catalase-polyethylene glycol, Ciclopirox olamine, Deferoxamine, diphenyleneiodonium chloride, DL-Dithiothreitol, Ferrostatin-1 (ChemBridge Corporation), Hydroxyurea, N-Acetyl-L-Cysteine, Necrostatin-1 (inactive) (Merck), (±)-α-Tocopherol, Z-Val-Ala-Asp(OMe)-CH2F (zVAD-FMK; Peptitde Institute), (+)-JQ1.

**RNA analysis by real-time quantitative PCR**. Total RNA was extracted using TRIzol (Life Technologies), followed by reverse transcription using GoScript Reverse Transcriptase (Promega) according to the manufacturer's instructions. Real-time quantitative PCR (RT-PCR) was performed using Brilliant SYBR Green

(Stratagene) on an i-Cycler (Bio-Rad Laboratories) according to manufacturer's protocol. Expression was normalized to the housekeeping gene *Tbp* for mRNA expression, or to genomic *Txnrd1* for addressing DNA recombination efficiency in *Txnrd1*-deficient cells. The sequences of all used primers are listed in Supplementary Table 3.

**Immunoblotting**. Purified splenic T cells were lysed on ice with RIPA buffer (20 mM Tris-HCl, pH 7.5, 150 mM NaCl, 5 mM EDTA, 1 mM Na$_3$VO$_4$, 1% Triton X-100, supplemented with protease inhibitor [Sigma-Aldrich] and phosphatase inhibitor [Sigma-Aldrich]). Samples were then spun for 10 min at 4 °C to remove all cell debris. Protein concentrations were determined using the *Pierce*$^{TM}$ BCA Protein Assay Kit (Thermo Scientific). Whole cell extracts (30 μg of proteins) were fractionated by SDS-PAGE and transferred to a Polyvinylidene difluoride (PVDF) membrane using a transfer apparatus according to manufacturer's instructions (Bio-Rad). After blocking with 4% nonfat milk in TBST (50 mM Tris, pH 8.0, 150 mM NaCl, 0.1% Tween 20) for 45 min, the membrane was washed once with TBST and incubated with primary antibodies (1:1000 in TBST with 4% BSA) at 4 °C for 12 h. Membranes were washed three times for 10 min and incubated with a 1:2000 dilution (in TBST with 4% nonfat milk) of horseradish peroxidase-conjugated anti-rabbit antibody for 1 h. Blots were washed with TBST three times and developed with the ECL system (Thermo Scientific) according to manufacturer's instructions. Antibodies for phospho-AMPKα (Thr172; 40H9; Cell Signalling Technology) and β-Actin (AC-15; Sigma-Aldrich) were used in this study. The uncropped immunoblots are shown in Supplementary Fig. 14.

**RNA preparation and sequencing**. Mice were injected with 100 μg Staphylococcal enterotoxin B i.p. (Sigma-Aldrich; or PBS as control). Vβ8.1/8.2$^+$ CD8$^+$ and total CD8$^+$ T cells were FACS-sorted from SEB- and PBS-injected mice, respectively. Approximately 300,000 sorted cells were resuspended in TRIzol (life technologies), phase separation was achieved by addition of chloroform (Sigma-Aldrich) and RNA was precipitated from the aqueous layer with isopropanol (Sigma-Aldrich) using glycogen (Roche) as a carrier. The TruSeq RNA Stranded Sample Prep kit (Illumina) was used to construct the sequencing libraries. In brief, total RNA samples (100 ng) were poly A enriched and reverse transcribed into double-stranded cDNA. TruSeq adapters were then ligated to double-stranded cDNA. Fragments containing TruSeq adapters on both ends were selectively enriched with PCR and subsequently sequenced on the Illumina NextSeq 500 in single-end mode, 150 cycles at the Functional Genomics Center Zurich (FGCZ). The fragments were mapped to the ensemble mouse reference genome GRCm38 (Version25.06.2015) using the STAR aligner[67]. For normalization, the read counts were scaled with the trimmed mean of *M*-values (TMM) method proposed by Robinson and Oshlack[68]. For differential expression analysis, the Bioconductor package edgeR was used. For the generation of the heat maps, genes with a |generalized fold change (GFOLD)| larger than 1.5 and a false-discovery rate of <0.2 were chosen for visualization. For the analysis of genes involved in the cell cycle, the KEGG cell cycle pathway (mmu04110) was used.

**Extracellular flux analysis**. Extracellular flux assay was performed with a Seahorse XF-96 Extracellular Flux Analyzer (Seahorse Bioscience) according to protocols described elsewhere[69]. Oxygen consumption rates (OCR) and extracellular acid-ification rates (ECAR) were measured in unbuffered RPMI 1640 with 2 mM L-glutamine containing 25 mM or without glucose. Baseline levels and responses to the indicated compounds were determined. To address mitochondrial stress, 1 μM oligomycin, 1.5 μM FCCP, and 500 nM rotenone + 1 μM antimycin A were provided (all from Seahorse Bioscience). For glycolysis stress analysis, 25 mM glucose (Sigma-Aldrich), 1 μM oligomycin, and 45 mM 2′-deoxy-D-glucose (Sigma-Aldrich) were added.

**Metabolomics**. Freshly-isolated, MACS-sorted (with CD90.2 microbeads) T cells were either analyzed directly after isolation or at different time points after αCD3/CD28 stimulation. Approximately 500,000 cells were washed twice with PBS and snap frozen in liquid nitrogen. Intracellular metabolites were extracted twice with cold (−20 °C) 40:40:20 acetonitrile:methanol:water. Non-targeted metabolite pro-filing was performed by flow injection analysis on an Agilent 6550 QTOF instrument in negative mode as previously reported[32]. Ions were directly annotated to metabolites based on the measured mass using the HMDM v3.0 database[32,33]. For targeted metabolomics, intracellular nucleotides were extracted with the same procedure, dried in a speed-vac and resuspended in 60 μl of ddH$_2$O. Relative levels of XMP and dXTP were quantified by ion-pairing LC–MS/MS in negative mode with multiple reaction monitoring using the MRM settings listed in Supplementary Table 4. The data were analyzed as previously described[70].

**Statistical analysis**. Two-group comparisons were assessed with a Student's *t* test (two-tailed, unpaired). One-sample *t* test was used to compare the mean of a group with specified value. Multi-group comparisons were assessed by one-way ANOVA followed by either Tukey's or Dunnett's corrections. For metabolomics data, the *p*-values were adjusted using the Benjamini–Hochberg multiple testing correction. The data are represented as mean + standard deviation, and the method of sta-tistical evaluation and the significance levels are indicated in each figure legend.

**Data availability**. Sequence data that support the findings of this study have been deposited in GEO with the primary accession code GSE107090. The authors declare that all the other data supporting the findings of this study are available within the article and its supplementary information files and from the corresponding authors upon reasonable request.

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

## Acknowledgements

We thank Christian Wolfrum and Miroslav Baláz for help in using the Seahorse instrument; Wilhelm Krek and Andrea Aloia for sharing pAMPK and pATR antibodies; Gianluca Figlia and Georg Bornkamm for discussions about AMPK activation and the thioredoxin system, respectively; Anette Schütz and Malgorzata Kisielow at the ETH Flow Cytometry Core Facility for cell sorting. We acknowledge the use of the Immgen database as an informative tool for our study. We are grateful for research grants from SNF (310030_163443/1) and ETH Zurich (ETH-23-16-2).

## Author contributions

J.M. and S.H. designed, performed, and analyzed the majority of experiments. M.M., L.P. and T.F. contributed to important methods and technologies. M.C. provided the *Txnrd1*^fl/fl mice. L.T., J.K., N.Z. and M.K. analyzed data and conceptualization. J.M. and M.K. wrote the manuscript.

## Additional information

**Competing interests:** The authors declare no competing interests.

