## [Peer Review File · Nature Communications]

Reviewers' comments:

Reviewer #1 (Remarks to the Author):

I thank the authors for addressing the comments, in present form paper is acceptable for publication with small explicit addition of GSE number to the appropriate place in the methods section.

Reviewer #2 (Remarks to the Author):

The authors have addressed most of my concerns satisfactorily. However, a couple still remain.

Remaining concerns:

Figure 1: The earliest iNKT cell subset (stage 0 iNKT cells) is present at a very low frequency in the thymus, in the region of 500 cells per thymus. The fact that the authors show data for 10,000 stage 0 iNKT cells makes me question the gating strategy. The authors characterize stage 0 cells as pre-gated on TCR β intCD1d-PBS57-tetramer+ and then CD24hiCD44loNK1.1-. However, any contamination of the TCR β intCD1d-PBS57-tetramer+ gate with conventional T cells will give contaminating CD24hiCD44loNK1.1- cell. To confidently gate on stage 0 cells requires the identification of PLZF+ thymocytes; PLZF+ is expressed by developing iNKT cells but not conventional T cells.

Figure 5B: AMPK data is not convincing. pAMPK T172 is highly subject to artefact (according to the AMPK expert – Prof. Hardie). Increased AMPK activity should be confirmed by measuring the phosphorylation of the AMPK substrate Acetyl-CoA Carboxylase on Serine 79. If AMPK is turned on after 2 days, is this due to energy crisis, ie decreased ATP levels? Given that Thioredoxin-reductase is important in the anti-oxidant response, I would have thought a likely explanation for the observed increased glycolysis might be increased levels of ROS leading to increased HIF1a stability and increased expression of glycolytic genes such as Glut1, Hex2, Ldha.

Figure 7: pH2AX can result from both DNA damage due to stalled replication forks and also due to direct DNA damages due to, for instance, ROS. The reason that I suggested that the authors look at pATR is because that is typically the DNA damage sensor that is activated due to stalled replication forks with pATM associated with DNA damage (strand breaks) due to ROS. The additional data provided does not further support the argument that the DNA damage is due to stalled replication forks.

Minor points:

Supplementary Figure 1B appears to show Txnrd DNA in the KO cells but not the controls. Is this an error?

Figure 2E: The y-axis label is unclear to me. % EdU+ (of CD8+). DN4 thymocytes do not express

CD8.

Point-by-point response for MS # NCOMMS-17-33522A

Reviewers' comments:

Reviewer #1 (Remarks to the Author):

I thank the authors for addressing the comments, in present form paper is acceptable for publication with small explicit addition of GSE number to the appropriate place in the methods section.

We thank the reviewer for her/his support of our study. As requested, we have now added the GSE number in the method section (see subsection "data availability").

Reviewer #2 (Remarks to the Author):

The authors have addressed most of my concerns satisfactorily. However, a couple still remain.

We thank the reviewer for carefully reading our point-by-point reply. Here we have addressed and discussed her/his remaining concerns.

Remaining concerns:

1) Figure 1: The earliest iNKT cell subset (stage 0 iNKT cells) is present at a very low frequency in the thymus, in the region of 500 cells per thymus. The fact that the authors show data for 10,000 stage 0 iNKT cells makes me question the gating strategy. The authors characterize stage 0 cells as pre-gated on TCR β intCD1d-PBS57-tetramer+ and then CD24hiCD44loNK1.1-. However, any contamination of the TCR β intCD1d-PBS57-tetramer+ gate with conventional T cells will give contaminating CD24hiCD44loNK1.1- cell. To confidently gate on stage 0 cells requires the identification of PLZF+ thymocytes; PLZF+ is expressed by developing iNKT cells but not conventional T cells.

We thank the reviewer for sharing her/his expertise in the field, and we agree that any contamination of the TCR β intCD1d-PBS57-tetramer+ gate with conventional T cells will give contaminating CD24hiCD44loNK1.1- cells. As suggested, we have now used the PLZF marker to confidently identify stage 0 iNKTs. To further minimize possible contaminations in the CD1d-PBS57-tetramer+ gate, we have additionally MACS-enriched on the basis of CD1d-PBS57-tetramer-APC and anti-APC beads. This has allowed us to work with a much cleaner CD1d-PBS57-tetramer+ gate. As shown in the new Figure 1g, we have now obtained ca. 400 cells per thymus for the earliest iNKT subset (stage 0), and notably this subset was present in the same numbers in WT and KO cells. By contrast, stage 1, 2 and 3 iNKTs were reduced in the absence of the Trx system, thereby confirming a defect in expansion rather than failed thymic selection.

2) Figure 5B: AMPK data is not convincing. pAMPK T172 is highly subject to artefact (according to the AMPK expert – Prof. Hardie). Increased AMPK activity should be confirmed by measuring the phosphorylation of the AMPK substrate Acetyl-CoA Carboxylase on Serine 79. If AMPK is turned on after 2 days, is this due to energy crisis, ie decreased ATP levels? Given that Thioredoxin-reductase is important in the anti-oxidant response, I would have thought a likely explanation for the observed increased glycolysis might be increased levels of ROS leading to increased HIF1a stability and increased expression of glycolytic genes such as Glut1, Hex2, Ldha.

Indeed, Trx system is known to play a crucial role in the antioxidant response. Furthermore, the GSH system is also a critical player in maintaining ROS homeostasis. Both systems have many overlapping functions and are known to compensate for each other functions in a cell type-specific manner^{1,2}. Our data indicate that the GSH system is able to maintain ROS homeostasis in the absence of the Trx system in T cells; indeed, we have not observed an increase in ROS levels in *Txnrd1*-deficient T cells upon activation (Supplementary Figure 4G) and the impaired proliferative capacity in the absence of *Txnrd1* was not improved by supplementation of well-known antioxidants (Figure 4K). Moreover, as described in Fig. 6c and Supplementary Fig. 6c-d, we have observed increased expression of *Srxn1* and *Nqo1* in *Txnrd1*-deficient T cells, indicating that these other factors might take over the antioxidant function of the Trx system. Taken together, these data exclude higher ROS levels in KO. Thus, the hypothesis that increased glycolysis might be due to ROS-mediated HIF1a stability is implausible.

To provide more evidence of AMPK activation to the reviewer, we have now included in the revised version of the manuscript the ATP-to-AMP ratio in WT and KO cells after aCD3/aCD28 stimulation (see new Figure 5c). As depicted, the ATP-to-AMP ratio is lower in the absence of the Trx system, in line with the observed higher phosphorylation of AMPK in the KO. Therefore, these data indicate that AMPK activation in the absence of the Trx system is caused by energy crisis (as it was suggested by the reviewer).

We hope that the reviewer will now agree that the higher metabolic activity in terms of glycolysis and OXPHOS observed with Seahorse measurements in the absence of TrxR1 is due to increased phosphorylation of AMPK caused by energy crisis.

3) Figure 7: pH2AX can result from both DNA damage due to stalled replication forks and also due to direct DNA damages due to, for instance, ROS. The reason that I suggested that the authors look at pATR is because that is typically the DNA damage sensor that is activated due to stalled replication forks with pATM associated with DNA damage (strand breaks) due to ROS. The additional data provided does not further support the argument that the DNA damage is due to stalled replication forks.

As we have extensively discussed in point 2), our data strongly exclude an increase in ROS levels in the absence of the Trx system, thereby suggesting that phosphorylation of H2AX is very likely due to stalled replication forks rather than direct DNA damage due to ROS.

To support this argument, we have followed the reviewer's suggestion and have measured phosphorylation of the ATR sensor protein. As expected, we observed increased pATR levels upon aCD3/aCD28 stimulation in *Txnrd1*-deficient cells (see new Supplementary Figure 7c), in line with replication stress due to reduced availability of DNA building blocks leading to H2AX phosphorylation.

Minor points:

Supplementary Figure 1B appears to show Txnrd DNA in the KO cells but not the controls. Is this an error?

We are sorry about this misunderstanding. As it was described in the figure legend, we had measured Cre-loxP recombination at the gDNA level by real-time PCR. This was achieved by using primers that only give a PCR product when the exon 15 of *Txnrd1* is deleted, therefore only in KO cells.

However, to render this clearer for the readers, we have now replaced this panel and showed the unrecombined *Txnrd1* DNA (see new Supplementary Figure 1B and Supplementary Figure 2B), which is expected to be present only in WT cells.

Figure 2E: The y-axis label is unclear to me. % EdU+ (of CD8+). DN4 thymocytes do not express CD8.

We thank the reviewer for highlighting this error. The y-axis label is now changed in "% EdU+ (of DN4)".

References:

1. Lillig, C.H. & Holmgren, A. Thioredoxin and related molecules--from biology to health and disease. *Antioxid Redox Signal* **9**, 25-47 (2007).
2. Holmgren, A. Thioredoxin and glutaredoxin systems. *J Biol Chem* **264**, 13963-13966 (1989).

REVIEWERS' COMMENTS:

Reviewer #2 (Remarks to the Author):

The authors have satisfactorily addressed all my concerns. I would like to congratulate the authors on a very nice study.